# Abundance-biased codon diversification prevents recombination in AAV production and ensures robust in vivo expression of functional FRET sensors

Jan Dernic [1,2,6], Afroditi Eleftheriou [1,2,6], Lazaros Vasilikos [3], Melanie Rauch[3], Pascal Imseng[1,2], Henri S. Zanker [1,2], Zoe J. Looser[1,2], Rachel M. Meister [1,2], Felipe Velasquez Moros [1,2], Tomer Kagan [4], Tal Laviv [4,5], Jean-Charles Paterna[3], Michael Arand [1], Aiman S. Saab [1,2], Bruno Weber [1,2] & Luca Ravotto [1,2] ✉

The delivery of genetically encoded fluorescent sensors via adeno-associated viral vectors (AAVs) enables the quantification of biological analytes with high spatiotemporal resolution in living animals. In this study, we expose an unreported problem of the approach, in which the presence of repeated subsequences in the sensor's DNA sequence triggers recombination during AAV production. In the case of Förster Resonance Energy Transfer (FRET) sensors, recombination leads to a mixture of fluorescent products, severely compromising in vivo functionality. To counter this phenomenon, we introduce Abundance-Biased Codon Diversification (ABCD), a modification of a previously reported codon diversification method that prevents recombination without sacrificing codon optimization for a target organism. We demonstrate that ABCD greatly facilitates in vivo studies by restoring the functionality of FRET sensors and advanced inducible expression systems delivered via AAV vectors. Our approach offers a robust solution to a previously overlooked challenge, significantly expanding the range of future applications in quantitative imaging and genetic manipulation in living animals using AAV-mediated strategies.

Adeno-associated viral vectors (AAVs) are the tool of choice for in vivo gene delivery due to their versatility, low immunotoxicity, and lack of genome integration, with applications ranging from therapeutical approaches in humans[1,2] to long-term expression of exogenous proteins in animals[3–5].

In particular, the AAV-mediated delivery of genetically encoded sensors (GESs) based on fluorescent proteins (FPs) has provided a minimally invasive way of monitoring biological processes with subcellular and sub-second resolution in living animals[6,7]. Among GESs, those based on Förster Resonance Energy Transfer (FRET) are a key tool for quantitative imaging, achieved through ratiometric or lifetime (FLIM) detection[8]. The first FRET GES, Cameleon[9], reported calcium concentration changes using the CFP-YFP fluorescent protein pair. In the following two decades, the FRET

principle was used to design sensors for a vast number of analytes, using different protein pairs or protein-dye constructs[6,10]. Despite the variety of approaches, the CFP-YFP pair has been used in the vast majority (>90%) of FRET sensors, likely due to the development of cyan and yellow FPs with excellent photophysical and chemical properties[11–14] and to the brightness, photostability and maturation issues of early generations of red FPs[15,16]. Quite surprisingly, despite the great potential of FRET and FLIM for in vivo imaging[8], a relatively low number of studies have used FRET biosensors in living animals[10,17]. Most of those studies relied on transgenic animals or in-utero electroporation, with only a few sensors delivered via AAVs to the best of our knowledge (Laconic and Pyronic for lactate and pyruvate[18,19], ATeam for ATP[20–22], and the AKAR family of sensor for protein kinase A[23,24]). This observation is in striking contrast with the widespread usage of AAVs for the

[1]Institute of Pharmacology and Toxicology, University of Zurich, Zurich, Switzerland. [2]Neuroscience Center Zurich, University and ETH Zurich, Zurich, Switzerland. [3]Viral Vector Facility, University of Zurich and Swiss Federal Institute (ETH) Zurich, Zurich, Switzerland. [4]Department of Physiology and Pharmacology, Gray Faculty of Medical and Health Sciences, Tel Aviv University, Tel Aviv, Israel. [5]Sagol School of Neuroscience, Tel Aviv University, Tel Aviv, Israel. [6]These authors contributed equally: Jan Dernic, Afroditi Eleftheriou. ✉e-mail: luca.ravotto@pharma.uzh.ch

convenient delivery of single fluorophore sensors in living animals (most notably, the GCaMP family of calcium sensors).

In this study, we reveal a previously unreported recombination process occurring during the production of AAV vectors that contain repeated DNA sequences. For FRET sensors based on the CFP-YFP pair, recombination prevents consistent and reliable quantitative imaging in living animals, hindering the adoption of this technology for in vivo studies. As a countermeasure, we modified a combinatorial codon scrambling (CCS) algorithm originally developed for PCR amplification of repetitive DNA sequences, introducing the concept of Abundance-Biased Codon Diversification (ABCD). The ABCD approach integrates a bias for organism-specific codon usage, simultaneously reducing potential concerns of diminished expression levels and upholding the diversification power of CCS. Our approach ensures a very high degree of codon diversification regardless of the chosen abundance bias, preventing recombination in AAV vectors. This is especially evident in complex cases that involve three fluorescent proteins within a single construct. Using the ABCD approach, we restored the functionality of a FRET sensor for glucose expressed in the mouse brain cortex, without affecting its physicochemical properties. To demonstrate the utility of our solution beyond the field of genetically encoded sensors, we addressed the recombination challenge in an inducible Cre recombination system that features dual hERT2 ligand binding domains, designed to combat background recombination activity in classical Cre-hERT2 constructs[25,26]. In essence, our ABCD strategy represents a comprehensive solution to a severe issue spanning the entire domain of AAV-mediated genetic manipulation in living organisms.

## Results

### Recombination occurs during the preparation of AAV vectors encoding constructs containing repeated sequences

During the preparation of AAV vectors encoding the glucose sensor FLII¹²P-700μδ6 (FLIIP)[27] and the ATP sensor ATeam1.03 (ATeam)[28], we performed routine quality checks of the viral preparations using PCR amplification and gel electrophoresis.

The expected PCR fragment length should closely match the sequence length of the respective sensor. However, for both FLIIP and ATeam, we observed a band indicating a loss of approximately 1600 and 1300 base pairs (bp), respectively (Fig. 1a). Upon expression of FLIIP in the adult mouse cortex through intracortical AAV delivery and ratiometric two-photon imaging, we detected the presence of fluorescence from the nucleus (Fig. 1b, left). This behavior is in stark contrast with what we previously observed with the lactate FRET sensor Laconic[29] and intriguingly with the ATP sensor ATeam1.03YEMK (Fig. 1b, right), which differs from the variant used in this study only by a few mutations in the binding pocket that increase the affinity for ATP[28]. Sanger sequencing of the PCR preparations (Fig. 1c, SI1, SI2) revealed that the viral vectors contained a mixture of fluorescent proteins instead of the expected sensors, as evidenced by the overlap of nucleotide signals in the (few) regions in which the sequences of the donor and acceptor FPs are different. Since the plasmids used for the viral preparations were found to contain the correct sequences (Fig. SI3), recombination could have occurred during PCR amplification and/or AAV preparation. Indeed, PCR amplification of a non-recombined plasmid containing the FLIIP sequence leads to the presence of recombined products following gel electrophoresis (Fig. SI4a). To assess the possibility of recombination occurring also during AAV production, original AAV samples (without amplification) were analyzed by gel electrophoresis. For both FLIIP (Fig. SI4a) and ATeam (Fig. SI4e), recombination bands were detected. By performing next-generation sequencing on the recombined constructs, we confirmed the presence of individual sequences combining CFP and YFP subsequences (Fig. SI4c, f). Furthermore, our data show that the recombination probability grows linearly as a function of the nucleotide position (Fig. SI4d, g), suggesting that, at least in our experimental set-up, recombination occurs with approximately equal frequency at any position.

The presence of a mixture of FPs was previously observed in cells transduced using lentiviral preparations[30], and interpreted as recombination, which requires the presence of repeated sequences with high level of similarity. This explanation is consistent with the fact that Laconic preparations exclusively contain the complete sensor sequence. In fact, Laconic contains the Clavularia sp.-derived mTFP as donor and the A. Victoria-derived Venus as acceptor, and their sequences are substantially different, preventing recombination. The case of ATeam, featuring a circularly permuted cp173Venus as acceptor, is particularly interesting. While sequencing (Figs. SI2, SI4) indicates that recombination occurs, the sensor has been previously shown to be fully functional in vivo in a direct comparison between a transgenic mouse line and an AAV delivery approach[21]. For ATeam, the circular permutation of the acceptor FP implies two possible sets of recombination products (Fig. 1d): "truncated FPs" that contain only the amino acids from 1 to 173, and "tandem FPs", where a complete FP is joined to a truncated one. In our case, gel electrophoresis and DNA sequencing suggest that the main recombination product are the truncated FPs, in which the missing part exceeds the maximum number of tolerated deletions[31], beyond which no fluorescence is possible. Consequently, for ATeam the only fluorescent product is the functional sensor (Fig. 1f), in accordance with previous reports of its successful AAV delivery.

### Combinatorial codon scrambling produces highly diversified DNA sequences using only species-specific abundant codons

Since sequence similarity is a prerequisite for the occurrence of recombination, we chose to address the problem using codon diversification, a procedure that generates a new sequence that is maximally diverse from a reference one at the nucleotide level, while retaining the same amino acid composition. To achieve this result, the codon scrambling algorithm introduced by Tang and Chilkoti[32] associates to each amino acid a list of all the codons that encode for it and then efficiently explores the space of codon combinations that produce a target amino acid sequence (Fig. 2a). The algorithm minimizes the value of an objective function ($f$) that depends on the interaction energies between subsequences, thus decreasing both sequence similarity and the tendency to form secondary RNA structures, which could potentially impact expression. The value of $f$ serves as a metric to compare different sequences, with lower values indicating a higher degree of diversification.

We modified this method by excluding those codons whose relative abundance in a target organism falls below a specified threshold (Fig. 2a). This strategy limits the number of sequences that can be explored, thus diminishing the diversification power, but reduces the risk of compromising the protein expression levels. The selected threshold level acts as a "weight factor" between the extremes of pure codon diversification and pure codon optimization. We label this approach as Abundance-Biased Codon Diversification (ABCD).

An important unknown of the procedure is the target $f$ value below which the output sequence is considered sufficiently diversified to prevent recombination. In their original paper, Tang and Chilkoti have found the $f$ value to span many orders of magnitude, and that sequences with an $f$ value $< 10^7$ were suitable for PCR amplification[32]. However, since primer binding in PCR and recombination in AAV production are different processes, the appropriate $f$ value threshold might also differ. To determine a suitable reference, we analyzed the original sequences of FLIIP, ATeam, and Laconic. Both FLIIP and ATeam show long identical stretches between the donor and acceptor FP (Fig. 1c, e), which lead to recombination. Their $f$ values are $> 10^{308}$ (the upper limit for double-precision floating point numbers) indicating an extremely high similarity, which the algorithm fails to quantify exactly. In principle, replacing the donor or acceptor FPs with more modern FP variants could reduce recombination, as newer variants carry additional mutations that reduce similarity. To investigate this idea, we have run similarity analysis for all combinations between the cyan variants mTurquoise2, Aquamarine, and mCerulean3 and the yellow variants Citrine, YPet, and SYFP2. For pairs involving YPet, the longest identical stretch was 72 bp, compared to all other cases in which stretches of more than 220 bp were

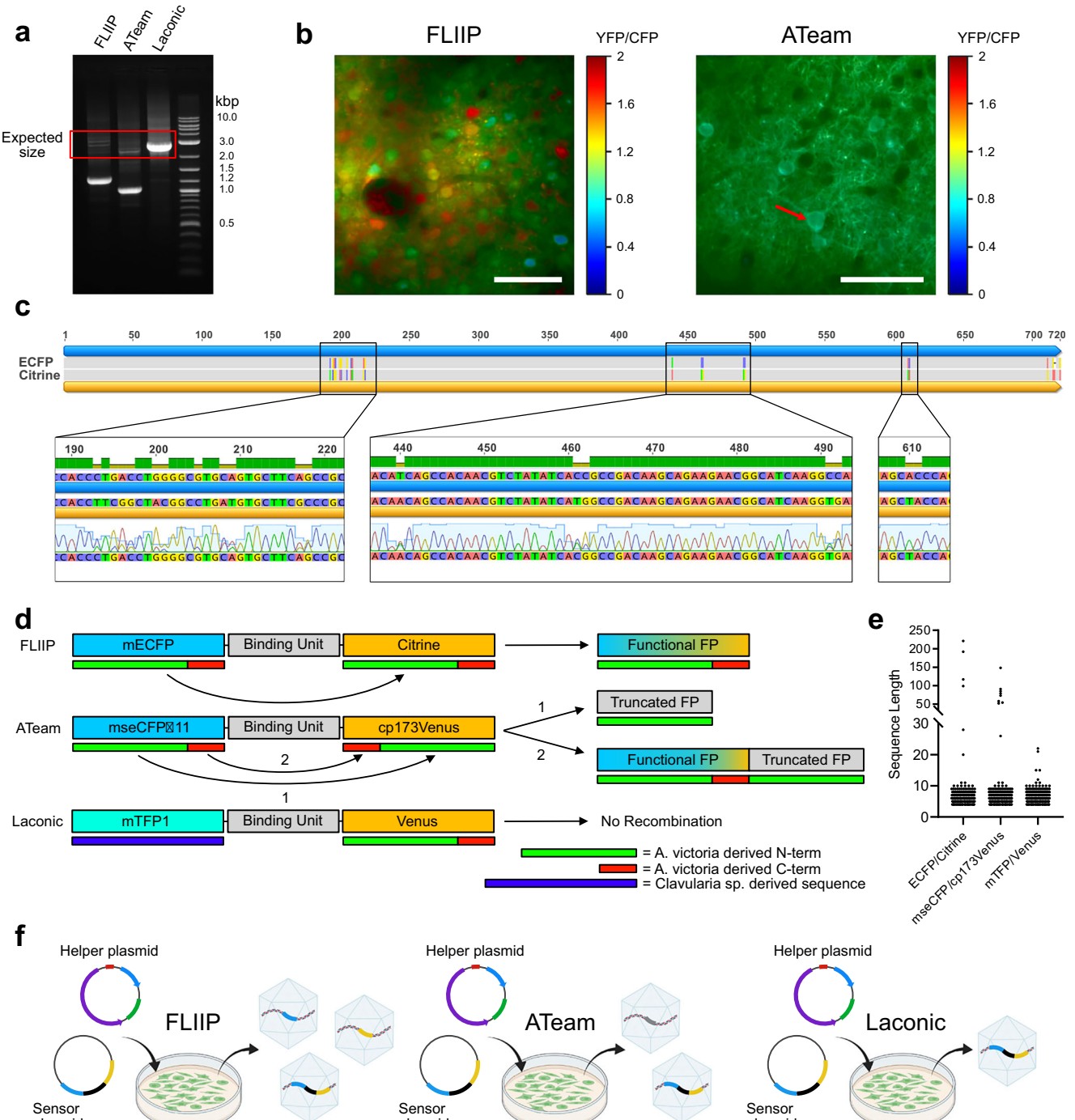

**Fig. 1 | Recombination in AAV vector preparations. a** Gel electrophoresis of the PCR-amplified viral vector DNAs for the three sensors, showing highly intense recombination bands for FLIIP and ATeam1.03. Primer binding sites and sequence lengths are reported in Figs. SI1, SI2. **b** Ratiometric two-photon microscopy images of neurons expressing FLIIP (AAV delivery) or ATeam1.03YEMK (transgenic mouse), showing the difference in nuclear localization. Scale bars = 40 μm. The red arrow highlights the presence of darker circular areas in the somata, corresponding to the cell nucleus. **c** Sequencing results of the PCR-amplified viral vector DNA for FLIIP. The presence of peaks belonging to two different nucleotides in positions in which the ECFP and Citrine sequences differ indicates the presence of a mix of FPs. **d** Schematic representation of the recombination possibilities for FLIIP, ATeam1.03, and Laconic. While for FLIIP recombination always results in a complete FP sequence, for ATeam there are two potential products, with only one of them being fluorescent. Color gradients indicate that recombination can occur at any point within the interval of nucleotides; **e** Subsequence homology length analysis for the donor-acceptor pairs of the three sensors, highlighting the presence of very long identical sequences between the donor and acceptor FP in both FLIIP and ATeam; **f** Schematic representation of the outcome of viral vector preparations for the three sensors. For FLIIP, viruses encoding both CFPs and YFPs are present together with a (minor) fraction of functional sensor. For ATeam, a large fraction of recombined FPs is also present, but its truncated sequence makes it non-fluorescent, allowing for the recording of the non-recombined sensor fraction. For Laconic there is no recombination and only the full sensor is present. Panel (**f**) was generated using BioRender.

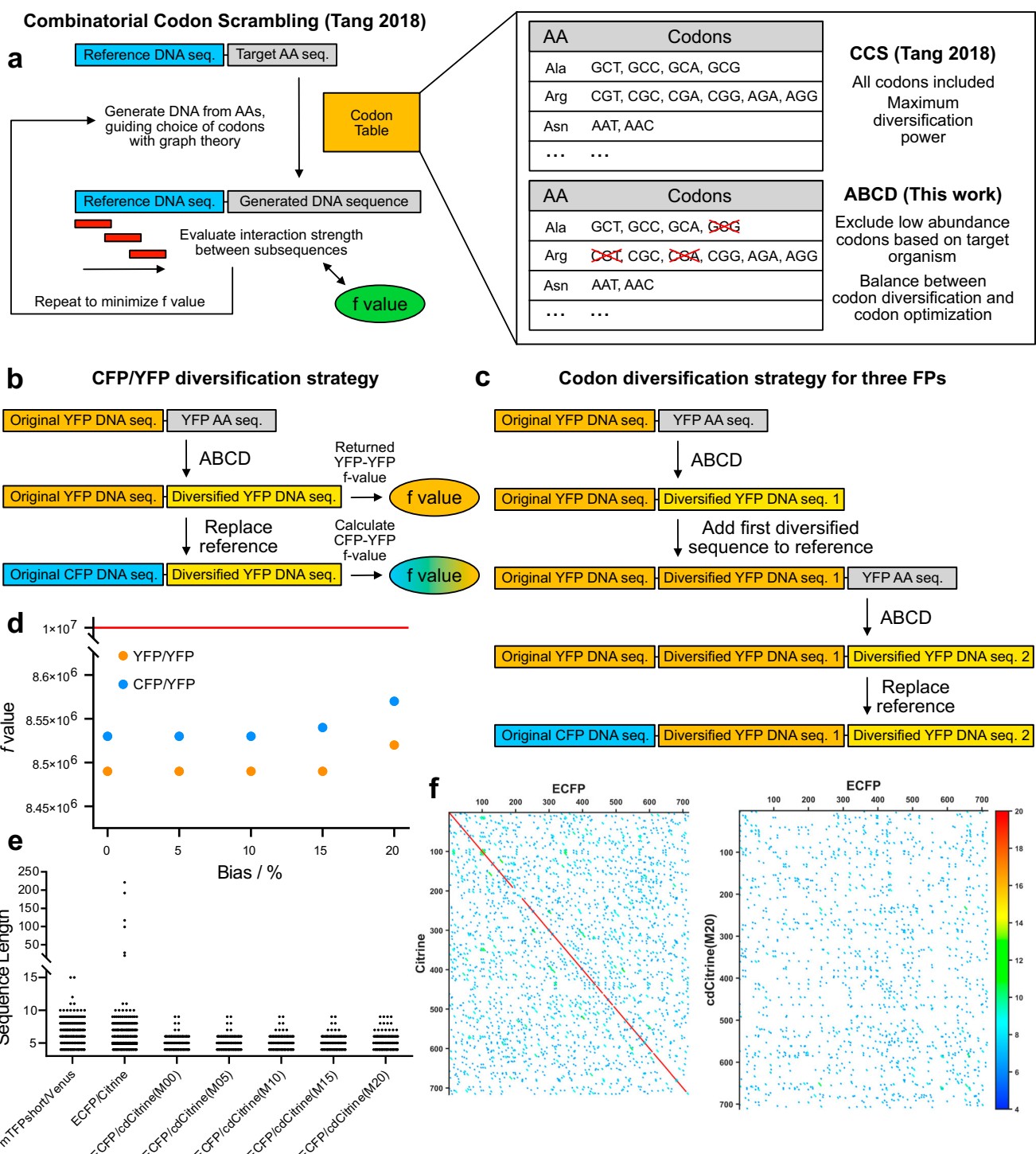

**Fig. 2 | Generation of diversified sensor sequences with the ABCD approach.**
**a** Schematic representation of the combinatorial codon scrambling algorithm proposed by Tang et al. and the abundance-biasing modification that defines the ABCD algorithm. The efficient exploration of the very large sequence space and the estimation of the interaction energy between subsequence pairs enable a very high degree of diversification, while the codon biasing prevents the usage of low-abundance codons, diminishing the chances of impacting expression. **b** Scheme of the algorithm used in this work to diversify FP FRET pairs, using the original YFP sequence as an internal reference to produce a new YFP sequence that is maximally diverse from it. The diversified YFP sequence is then compared with the CFP sequence of the corresponding FRET donor. **c** Algorithm scheme for the simultaneous diversification of three FPs. After a first round of diversification, the diversified sequence is included in the reference sequence for a second round of

diversification. **d** Obtained $f$ values for the Citrine diversification runs (yellow) and after substitution of the reference Citrine sequence with the ECFP one (blue). The impact of both the FP switch and the different codon abundance bias is negligible, and all $f$ values are below the approximative threshold for recombination observed by Tang et al. in their PCR study. **e** Distribution of identical subsequence length for different FRET pairs. While the original ECFP/Citrine pair contains very long identical stretches, all pairs containing a codon-diversified Citrine compare very favorably not only with the original pair, but also with the "naturally diversified" mTFP1/Venus pair. **f** Dot plots highlighting the presence of identical subsequences among FRET pairs. Codon diversification not only removes the very long stretches along the diagonal originating from the two FPs sharing a common ancestor, but also improves the diversity in the out-of-diagonal regions, corresponding to sequence comparisons involving a nucleotide shift between the donor and acceptor sequences.

identified. Nevertheless, in all cases the $f$ values were $> 10^{308}$, indicating that using more modern FPs is not a viable strategy to reduce recombination.

In contrast, the $f$ value for the non-recombining Laconic is $2.95 \times 10^{11}$. Interestingly, we noticed the presence of relatively long identical nucleotide stretches (Fig. 1e) at the N and C terminals (likely introduced during cloning), which are not essential for fluorescence. Omitting these sequences, the $f$ value decreased to $1.03 \times 10^{10}$. An $f$ value analysis of pairs of FPs coming from different organisms (Table SI1) shows values ranging from $\sim 10^{13}$ to $\sim 10^{5}$, with mNeonGreen and mStayGold appearing particularly diverse from other families of FPs, and among themselves. Thus, while higher $f$ values might be sufficient to avoid recombination (as shown in Laconic), as a conservative estimate, we consider the value of $\sim 10^7$ identified by Tang and Chilkoti in their PCR study a suitable target.

For both FLIIP and ATeam, we diversified the sequences of the YFPs while including the original sequence as a fixed N-terminal part, ensuring that the algorithm would minimize interactions both within the newly generated sequence and with respect to the one commonly used in sensors (Fig. 2a). Subsequently, we calculated the $f$ value for a pairing of the codon-diversified YFP (cdYFP) and its corresponding CFP, to better reflect the FP pair of each sensor (Fig. 2b). A more direct approach would have been to use the CFPs in the original diversification runs as the fixed N-terminal sequence, but we reasoned that diversifying a FP with respect to its own commonly used sequence would provide a more flexible strategy to replace FPs in existing sensors with their codon-diversified counterpart.

Finally, to demonstrate the applicability of the approach to more complex scenarios, we replaced the acceptor of ATeam1.03 with a tandem dimer of cp173Venus and Venus (cdATeamDA). A similar approach has been reported for the EPAC family of cAMP sensors as the only reliable strategy to eliminate aggregation in mammalian cells[33]. A double-acceptor strategy has also been shown to be effective in enhancing the FRET efficiency in FP-based systems[34]. This construct presents a particularly challenging problem for diversification since each FP must be maximally diverse from the other two simultaneously. In this case, we first applied the algorithm to cp173Venus, and then we added its diversified sequence to the fixed N-terminal part for the diversification of Venus (Fig. 2c). The result is a set of three FP sequences in which each one is maximally diverse from both of the other two. In all cases, by exploiting the capacity of the algorithm to avoid unwanted sequences, we excluded sets of restriction sites commonly used in our synthetic strategies and the viral inverted terminal repeats (ITRs), further minimizing the risk of recombination.

Using the described approach, we have produced diversified sequences of Citrine (cdCitrine), cp173Venus (cdcp173Venus), and the tandem dimer of cp173Venus and Venus (cdtdVenus) using abundance biases for mammalian expression between 0 and 20%. Analysis of the $f$ values (Fig. 2d, Table SI2) and of the length of identical subsequences (Fig. 2e, f) shows that nearly all diversification runs give $f$ value $< 10^7$, with a corresponding maximum length of identical subsequences $<10$ bp. Running the algorithm using the CFP donors as the fixed N-terminal sequence yields nearly identical $f$ values (Table SI1), validating our "self-diversification" idea.

Importantly, the effect of codon bias on the diversification efficiency appears to be marginal, demonstrating that the ABCD approach can reduce the risk of low expression without sacrificing diversification. However, this finding might not hold for more complex problems and higher biases, as demonstrated by cdtdVenus at 20% codon bias, which features a noticeably higher $f$ value and an identical subsequence of 17 bp. Additionally, the exclusion of critical sequences such as viral ITRs and relatively large sets of restriction sites (Tables SI3) has a negligible impact on the diversification efficiency, except again for cdtdVenus at 20% codon bias. In view of these results, we decided to synthesize sequences obtained at 20% bias for cdCitrine and 15% bias for cdtdVenus. It is worth noting that despite the fact that the combination of three FPs and 20% bias significantly increases the $f$ value, its absolute value of $\sim 10^8$ is still two orders of magnitude below the "natural diversification value" observed for Laconic.

## ABCD allows for robust in vivo quantitative imaging and delivery of advanced inducible expression systems

To generate codon-diversified GESs, we replaced the original acceptors in the GESs with their corresponding codon-diversified version and generated AAV vectors for neuronal expression.

In contrast with what we observed for the original FLIIP and ATeam, for all codon-diversified sensors gel electrophoresis shows a single amplicon at the expected full-length size, a clear sign that recombination was eliminated (Fig. 3a, b). To confirm this result in a more quantitative way, we performed Southern blotting of viral DNAs from original and codon-diversified FLIIP and ATeam constructs (Fig. SI5). The original versions displayed a distinct additional restriction fragment that is substantially smaller than the expected fragment yet specifically hybridizes with the GFP-specific probe. The size of this additional fragment is in line with an elimination of the analyte-binding domain between the two fluorescent proteins, resulting in a mix of FPs (Figs. SI1, SI2). While the recombined fragment is completely absent in the codon-diversified variants, its abundance in the case of the original sequences is substantially higher than that of the correct fragment. Considering that the latter contains only half of the probe-hybridizing sequence, semi-quantitative analysis of the Southern blot signals reveals a degree of recombination of $\sim 75\%$ for FLIIP and $\sim 66\%$ for ATeam. For cdATeamDA, restriction digest followed by gel electrophoresis showed a single band, and the sensor was found to be functional in cell experiments with metabolic blockers (Fig. SI5), confirming the success of the diversification strategy.

To demonstrate that ABCD did not influence the sensor's physico-chemical properties, we transfected HEK293 cells with plasmids containing either FLIIP or cdFLIIP, since transfection is not affected by recombination. The cells were perfused first with a glucose-free buffer, to completely deplete intracellular glucose, and then with a buffer containing 25 mM glucose, to reach high levels of intracellular glucose. As expected, the two sensors responded identically (Fig. 3c).

To illustrate the impact of recombination in quantitative imaging, we expressed both FLIIP and cdFLIIP in the brain of adult mice via intracortical injection of the corresponding AAV preparations. FRET images were collected upon irradiation at 870 nm and 925 nm. While the first irradiation wavelength reflects the typical use of the sensor (excitation of ECFP while minimizing the direct excitation of Citrine), in the latter case both FPs are excited with similar efficiency.

Recombination is clearly observable in the higher cell-to-cell variability (Fig. 3d, e) and nuclear localization (Fig. 3d, f) in FLIIP. Both effects are due to the presence of a mixture of FPs, which confer the variability and are small enough to cross the nuclear membrane[35]. For cdFLIIP, switching the wavelength from 870 to 925 nm caused a change in average ratio but not an increased variability (Fig. 3e). On the other hand, the significant increase in variability observed for FLIIP is proof of the presence of multiple emissive species, featuring different excitation efficiency at different wavelengths. The lack of nuclear localization for cdFLIIP is clearly visible in the images as the dark round spot in the somata, a well-known morphological feature. To give a more quantitative estimate, we calculated the percentage of pixels within each soma in which the intensity is below a certain fraction of the mean intensity of the same area, showing a clear difference between the two sensors (Fig. 3f). Next, we compared the response of FLIIP and cdFLIIP in vivo, performing intravenous glucose injections in anesthetized mice (Fig. 3g). Upon glucose injection, cdFLIIP shows ratiometric changes compatible with the expected increase in glucose concentration[36]. However, as the curve before injection is not perfectly stable, we performed a control injection using saline. Indeed, a drift of the signal is observed, but the larger change upon glucose injection confirms the sensor's functionality. On the contrary, FLIIP did not respond appreciably to the glucose injection.

The impact of recombination was further examined using fluorescence lifetime imaging (Fig. 4a). FLIM is the most robust method to quantitatively estimate analyte concentrations, allowing for the direct comparison between lifetime values obtained in different experiments[8,37]. Recombination in FLIIP clearly affects the lifetime values at baseline due the presence of the CFP fraction. Since the "free" CFP fraction is not quenched by FRET (Fig. 4b), one

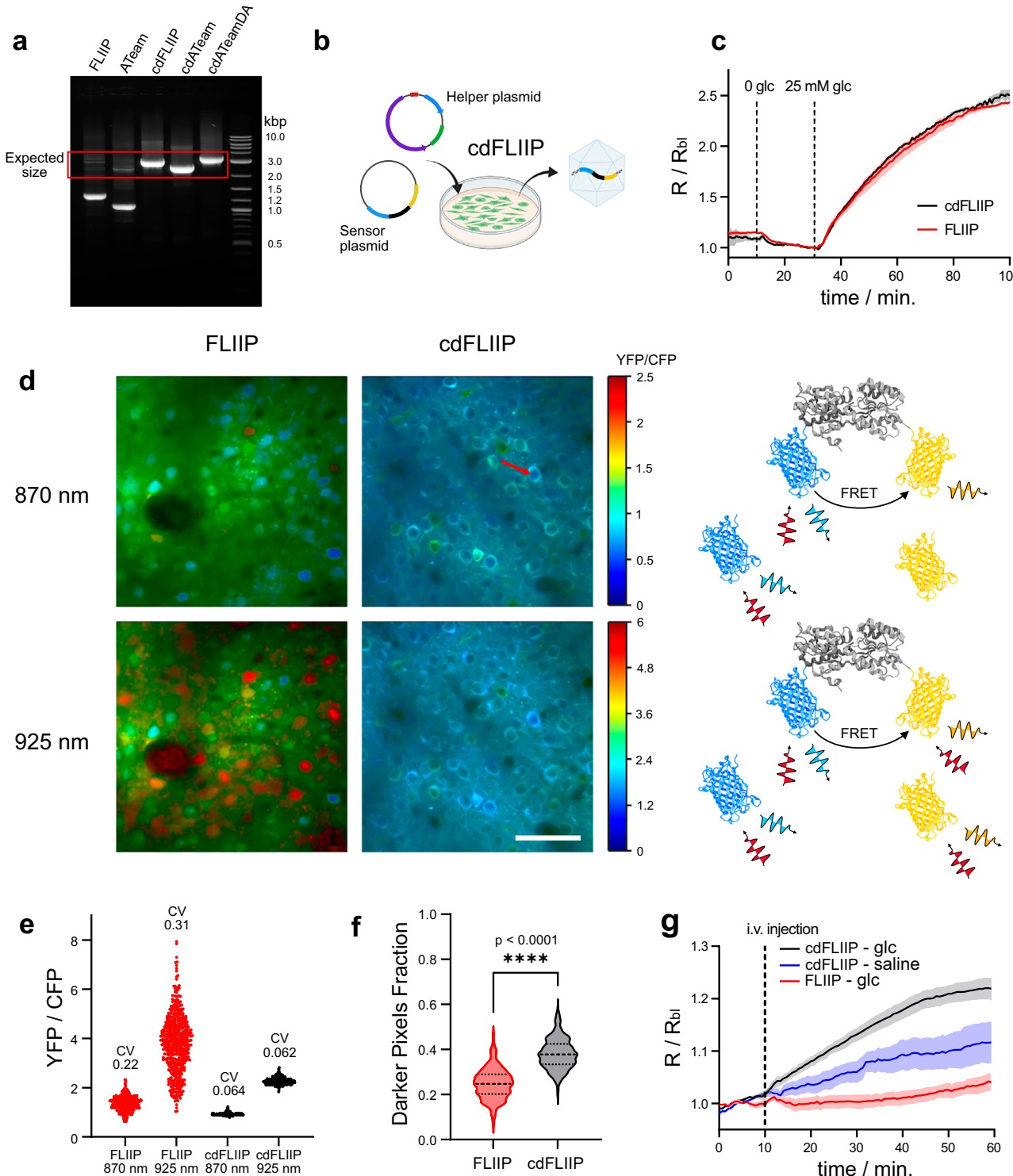

would expect FLIIP to show a bias towards longer lifetimes with respect to cdFLIIP. Instead, broad lifetime distributions are observed (Fig. 4c), with values both above and below those of cdFLIIP, likely due to the fact that recombination does not generate a single CFP but rather a distribution of structures with unknown effects on the photophysical properties. Upon glucose injection, the FLIM response (Fig. 4d) shows similar dynamics to what was observed for FRET (Fig. 3g), further confirming the functionality of the sensor.

A quantitative comparison of the expression levels between FLIIP and cdFLIIP is not straightforward due to the different nature of the emitting proteins in the two samples and the difficulty in quantifying intensity signals

in vivo. To obtain a qualitative estimate, we made sure to use identical viral titers, injection volumes, and instrumental parameters when two-photon microscopy experiments were performed. The signal for cdFLIIP was always comparable (if not higher) to that of FLIIP, confirming that ABCD did not lead to reduced expression.

To show how overcoming the recombination problem can greatly facilitate the use of FRET sensors in vivo, we aimed at expressing cdFLIIP in multiple brain areas upon intravenous injection of an AAV construct in adult mice. In absence of a viable AAV approach, generating such a large fluorescence distribution would require more cumbersome methodologies

**Fig. 3 | Elimination of recombination in FRET sensors and in vivo ratiometric imaging of AAV-delivered FLIIP and cdFLIIP. a** Gel electrophoresis of the PCR-amplified viral vector DNAs, showing highly intense recombination bands for FLIIP and ATeam and the absence of recombination for the corresponding codon-diversified versions (cdFLIIP and cdATeam), including a "double-acceptor" version of ATeam (cdATeamDA). **b** Schematic representation of the viral vector production outcome, showing that upon codon diversification only the full sensor sequence is present. **c** Ratiometric response of transfected HEK cells expressing FLIIP (red) or cdFLIIP (black) to bath glucose administration. Vertical bars indicate switching of the glucose concentration from 2 to 0 and from 0 to 25 mM. Since transfection is not affected by recombination, the two sensors behave identically. Curves are shown as mean ± SEM (FLIIP: $N = 28$ cells, $n = 3$ experiments; cdFLIIP: $N = 22$, $n = 3$). **d** Color-coded ratiometric images for cortical neurons expressing FLIIP or cdFLIIP, under 870 or 925 nm two-photon excitation. The schemes on the right represent the expected excitation and emission processes for the recombining FLIIP vectors. The fact that 925 nm light significantly excites YFPs directly allows better visualization of the cells containing predominantly YFPs. The red arrow highlights the presence of

darker circular areas in the somata, corresponding to the cell nucleus. Scale bar = 40 μm. **e** Distributions of the values of the CFP/YFP ratios for single cells. The coefficient of variation, computed as ratio of SD/mean, is reported for each distribution (means ± SD, n cells: FLIIP 870 nm = 1.36 ± 0.30, $n = 525$; FLIIP 925 nm = 3.79 ± 1.17, $n = 525$; cdFLIIP 870 nm = 0.92 ± 0.06, $n = 447$; cdFLIIP 925 nm = 2.27 ± 0.14, $n = 478$). **f** Fraction of pixels in each cell body whose intensity is below 90% of the intensity of the mean. The significant difference in this value reflects a significant difference in the distribution of the intensity values due to the darker nuclei. Dashed lines indicate the median, 25th and 75th percentile. The $p$ value was calculated based on Welch $t$ test. **g** Baseline normalized ratiometric responses of cortical neurons expressing FLIIP (red) or cdFLIIP (black) to a bolus i.v. glucose injection. As reference, the response of cdFLIIP to an identical bolus i.v. injection of saline is shown in blue. While cdFLIIP produces a response compatible with the expected rise in glucose concentration, FLIIP does not respond to the injection. Curves are shown as mean ± SEM ($N = 3$ mice, $n = 6$ experiments). Panels b and d were generated using BioRender.

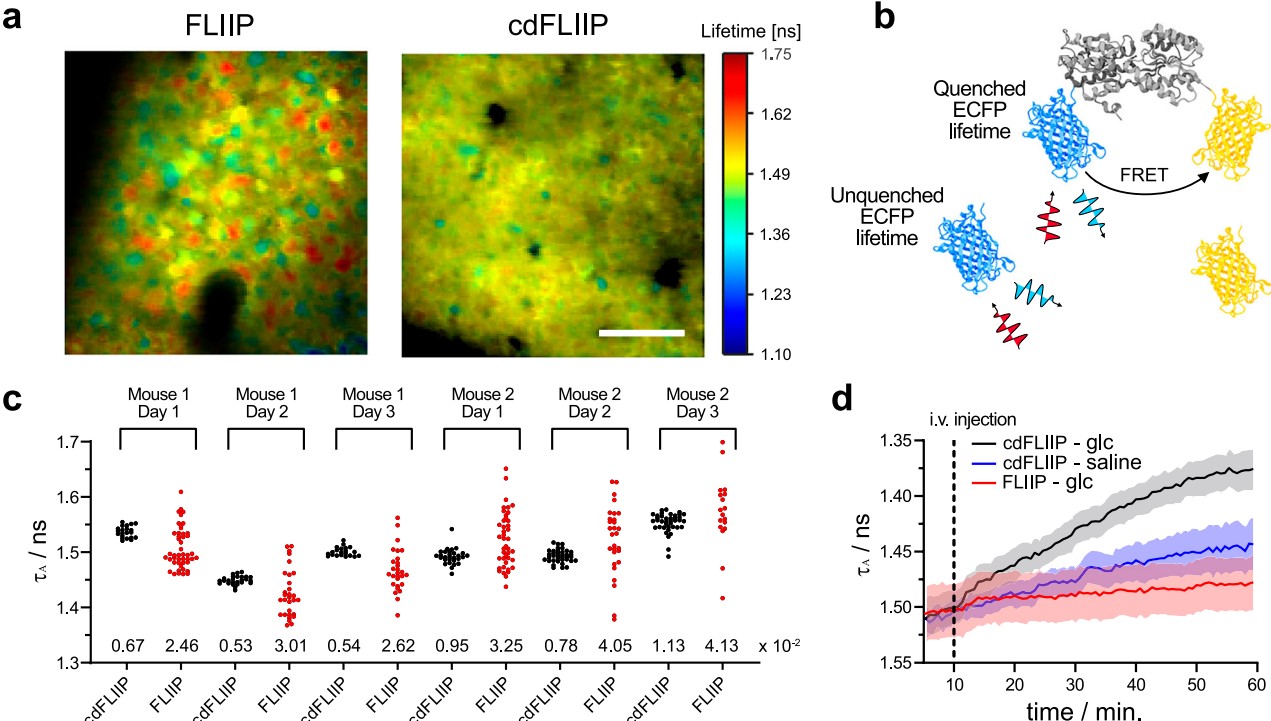

**Fig. 4 | In vivo FLIM imaging of AAV-delivered FLIIP and cdFLIIP. a** Color-coded FLIM images for cortical neurons expressing FLIIP or cdFLIIP under 870 nm two-photon excitation. Scale bar = 40 μm. **b** Schematic representation of the expected emissive species when recombination occurs. **c** Distribution of FLIM values in cells before glucose injection in (**d**), showing the reduction of variability associated with the prevention of recombination. For each experiment, the coefficient of

variation (mean/SD) is reported. **d** FLIM responses of cortical neurons expressing FLIIP (red) or cdFLIIP (black) to a bolus i.v. glucose injection, showing the lack of response as a result of recombination. As reference, the response of cdFLIIP to an identical bolus i.v. injection of saline is shown in blue. Curves are shown as mean ± SEM ($N = 3$ mice, $n = 6$ experiments). Panel (**b**) was generated using BioRender.

such as in-utero electroporation or the generation of transgenic lines. Including the cdFLIIP construct into the AAV-PHP.eB serotype, which is able to cross the blood-brain barrier, we were able to demonstrate for the first time the successful sparse expression of a FRET sensor across multiple brain regions via viral delivery (Fig. SI6).

The utility of the ABCD method is not limited to the field of genetically encoded fluorescent sensors. An exemplary application is the case of the hERT2-Cre-hERT2 inducible expression system. Contrary to the widely used Cre-hERT2, this construct presents an additional ligand-binding domain, which has been introduced to minimize the background expression of the Cre protein in absence of the inducing compound tamoxifen[25,26].

An initial attempt to produce an AAV vector resulted in constructs containing mainly the hERT2 sequence, with complete loss of the Cre protein (Fig. SI7). After discovering the recombination issue in FRET

sensors, we used the ABCD approach to produce a diversified sequence (cdhERT2) and the resulting construct, hERT2-iCre-cdhERT2, was successfully integrated into an AAV vector (Fig. SI7), providing a convenient method to address background expression for in vivo genetic studies.

## Discussion

The AAV-mediated expression of exogenous proteins is a widely employed technique to genetically manipulate organisms and study a vast range of biological processes in living animals, with high spatiotemporal resolution and cellular specificity. Here, we show that whenever a construct to be inserted into an AAV vector contains long repeated sequences, recombination can occur, generating a mixture of unwanted products. An exemplary case is that of FRET sensors based on the CFP-YFP pair. With the exceptions of mTFP[13] and the recently reported YFP Gamillus[38], virtually all

existing variants of CFP and YFP were evolved by mutagenesis from the original GFP sequence from A. Victoria[39]. As a result, the nucleotide sequences of the donor and acceptor proteins in most FRET sensors share sequence identities above 95%.

Komatsubara et al. have shown that this sequence similarity leads to recombination during the preparation of stable cell lines by lentiviral-mediated gene transfer[30]. As a result, cells express a mixture of fluorescent proteins in addition to the sensor, leading to high cell-to-cell variability and jeopardizing quantitative readouts. In order to decrease the level of sequence homology and avoid recombination, the acceptor protein was codon-optimized for expression in E.coli, while the donor was optimized for human expression. In a few other studies, FPs were also codon-diversified before lentiviral vector preparation, either using Komatsubara's method[40,41] or presumably by manual inspection of the DNA sequences[42-44]. Komatsubara's method can be considered a codon "deoptimization" procedure with respect to the target organism, since it relies on the choice of codons which are most abundant in a different one. However, codon usage is an important determinant of gene expression[45,46]. In particular, the importance of using abundant codons to enhance FP expression has been experimentally demonstrated in different organisms[47,48], starting from the initial "humanization" of the jellyfish-optimized sequence of GFP[49]. Thus, codon-deoptimizing an FP could result in suboptimal in vivo expression. Manual inspection might be used to exclude low-abundance codons depending on the target organism, but the procedure is tedious, and reaching the optimally diversified sequence is virtually impossible due to the daunting number of possible combinations. In fact, during the generation of tandem dimers of FPs derived from the same organism, manual diversification failed to completely solve the recombination problem[50]. Furthermore, cloning procedures require the absence of specific sequences in the gene of interest (e.g., restriction sites) and the complete redesign of a sequence carries the risk of formation of secondary structures at the mRNA level, which might affect translation. Previously, Aoki et al. used transposon-mediated gene transfer to avoid recombination in stable cell lines expressing a FRET sensor[51]. However, despite promising advances[52,53], transposon technology is not yet mature enough to compete with viral vectors for generating long-term, high-level expression in the brain, due to the need of complicated procedures such as in-utero electroporation[53] or the additional risks involved with genome integration[54]. A different approach would be the replacement of the donor CFP protein with the coral-derived mTFP, whose DNA sequence is substantially different from those of A. Victoria derived FPs. However, the physicochemical properties of the sensor might be affected, requiring further characterization and re-optimization. In fact, mTFP has been shown to dimerize inefficiently with A. Victoria derived YFPs, leading to sensors with reduced dynamic range[55].

Despite the difficulty of properly addressing the problem, the fact that FRET sensors have been successfully delivered using adenoviruses[56-58] or togaviruses[59] might have suggested that this form of recombination is a phenomenon limited to retroviruses, due to the involvement of reverse transcriptase[30]. Thus, our work unveils an unreported but very serious threat for studies involving the AAV-mediated delivery of genetic constructs.

Using two-photon microscopy in the brain cortex of living mice, we have shown that recombination severely affects both ratiometric and FLIM readouts of the glucose sensor FLIIP. The expression of multiple fluorescent proteins together with a non-recombined sensor fraction results in two main effects. First, due to the stochastic nature of viral gene delivery, every single cell will express a different mix of fluorescent species, giving rise to the spurious impression of a large cell-to-cell variability in the concentration of the target analyte. Second, since only the complete sensor responds to concentration changes, the recombined FP fractions produce, in the best scenario, an additional constant signal that reduces the dynamic range of the sensor. However, the problem is not limited to a reduced response, since a diminished range increases the weight of confounding factors (e.g. motion or hemodynamic effects), with no guarantee of a correct representation of the kinetic profile of the analyte. Recombination can be especially dangerous when comparing the analyte levels among different cell types, as it might

lead to interpreting differences in recombination efficiency as physiological differences in the concentration of the analyte.

During imaging, the effects of a high degree of recombination are immediately evident upon comparison with a non-recombining AAV preparation when analyzing the cell-to-cell variability. In absence of a non-recombining control, the expected degree of cell-to-cell variability in the analyte concentration is not known a priori, but a different degree of variability at different excitation wavelengths can be used as a clear indication of the presence of a mixture of emissive species. Nevertheless, as an imaging-independent method to identify recombination, we recommend routinely performing appropriate DNA analysis on viral vector preparations.

In the case of the ATP sensor ATeam1.03, circular permutation of the acceptor FP, a common design strategy used to optimize the dynamic range of FRET sensors[60], reduced the risk of data misinterpretation associated with recombination, implying "only" a reduction in the effective titer of the viral preparation and the expression of non-fluorescent partial FPs, with unknown physiological effects. The peculiar behavior of ATeam1.03 is consistent with the fact that it is one of the only two FRET sensors with a genetically related FP pair that has been successfully delivered via AAVs. The second case, AKAR-type sensors, also feature a circularly permuted acceptor[61]. However, sequence analysis predicts the possibility of formation of fluorescent "tandem dimers", containing a fully functional CFP linked to a non-fluorescent partial FP, depending on the position in the sequence where recombination occurs. In absence of information about the mechanism of recombination in AAV vectors, it is impossible to distinguish if the behavior of ATeam1.03 can be generalized to other sensors or whether is should rather be considered a "lucky accident", especially considering that sensors can differ in important factors such as the permutation position in the acceptor sequence or the switching of the position of donor and acceptor[62]. Ultimately, codon diversification is a much simpler and more practical solution than having to confirm the appropriate behavior of a recombining sensor and thus should be adopted in all cases.

A further demonstration of the utility of our methodology has been recently provided by the diversification of the phosphatase sensor G-PTEN[63], a FLIM-FRET sensor that features a dark acceptor (Fig. SI8). In this case, recombination is harder to recognize, as its product (an EGFP-like FP mix) has a similar lifetime to the low-FRET form of the sensor. However, the impact of recombination is clearly visible in the high-FRET form, which displays a much-reduced variability upon diversification.

Codon diversification can eliminate the recombination problem by modifying the DNA sequence without affecting the corresponding amino acid sequence. The inclusion of a codon abundance bias into a combinatorial algorithm for codon diversification provides an optimal solution. While in theory biasing reduces the efficiency of the original algorithm, in all practical cases its impact on diversification is negligible, yet it ensures that low-abundance codons, which could impair expression, are not present in the final sequence. The algorithm can be set to diversify an FP not only "internally" (excluding excessive similarity between subsequences of its own sequence) but also with respect to a target DNA sequence, such as the one commonly used in FP-based constructs. In this way, a library of diversified FPs can be generated and used to replace the corresponding FPs in FRET sensors. The risk of recombination of the diversified pairs is not only strikingly lower than the original pairs, as expected, but also significantly lower than the "naturally diversified" pair mTFP-Venus, which was used as a non-recombining control. The unique power of the ABCD approach was illustrated by the simultaneous abundance-biased codon diversification of three FPs, a problem that would be extremely difficult to solve efficiently with previous methods. The utility of delivering FRET sensors via AAV approaches was further proven by sparsely expressing cdFLIIP in multiple brain regions via i.v. injection of AAV vectors in adult mice. This convenient delivery route allows the quantification of biological processes across different brain areas without the need for dedicated mouse lines or surgical procedures such as in-utero electroporation. The ABCD approach is

applicable to any situation in which repeated sequences are delivered via AAV vectors, as demonstrated in the case of the hERT2-Cre-hERT2 inducible expression system, originally developed to reduce the "leakiness" of the Cre-hERT2 system. This idea was recently adopted as one of the key parameters for the implementation of an enhanced expression system capable of amplified Cre expression for the reliable generation of genetic mosaicism in mice[64], using in-utero electroporation. Thus, the possibility to reliably include the hERT2-Cre-hERT2 construct in an AAV vector might prove very advantageous for in vivo studies.

In conclusion, we introduced a robust solution to an unreported problem that can severely compromise studies based on AAV-delivered constructs. This work could foster the use of FRET sensors in vivo and help promote quantitative imaging and genetic approaches to tackle fundamental research questions in neuroscience and beyond.

## Methods
### Cloning
The codon-diversified sequences for cdCitrine(M20) and cdcp173tdVenus(M15) were custom-synthesized (ThermoFisher GeneArt) with inclusion of terminal or linking stretches containing restriction sites for insertion in plasmids for viral vector preparation (Supplementary Note 1). For the construction of the plasmids, the following procedures were employed (short names in parenthesis are identification codes of the UZH Viral Vector Facility inventory).

**pssAAV-2-hSyn1-Laconic-WPRE-hGHp(A) (pJS20).** The N-terminal part of the Laconic sequence was isolated by PCR from pssAAV-2-hGFAP-hHBbI/E-Laconic-bGHp(A) (kind gift of Sylvain Lengacher) with simultaneous introduction of a Kozak sequence, and reinserted into the original plasmid via NheI/KpnI restriction digestion and ligation. The entire Laconic sequence was then isolated by NheI/HindIII restriction digestion and inserted into HindIII/NheI restriction-digested pssAAV2-hSyn1-WPRE-hGHp(A).

**pssAAV-2-hSyn1-FLIIP-WPRE-hGHp(A) (pJS21).** The FLIIP sequence was isolated from pssAAV-hGFAP-hHBbI/E-FLIIP-bGHp(A) (kind gift of Sylvain Lengacher) by NheI/HindIII restriction digestion and inserted into HindIII/NheI restriction-digested pssAAV2-hSyn1-WPRE-hGHp(A).

**pssAAV-2-hCMV-chI-FLIIP-WPRE-SV40p(A) (p86).** The FLIIP sequence was isolated from pJS21 by NheI/HindIII restriction digestion and inserted into HindIII/SpeI restriction-digested pssAAV-2-hCMV-chI-EGFP-WPRE-SV40p(A).

**pssAAV-2-hCMV-chI-cdFLIIP(M20)-WPRE-bGHp(A) (p495).** The cdCitrine(M20) sequence was isolated by EcoRI/HindIII restriction digest from the synthesized sequence (Supplementary Note 1) and the N-terminal part of the FLIIP sensor was isolated by NheI/EcoRI digestion of pssAAV2-hSyn1-FLIIP-WPRE-hGHp(A). The two sequences were then inserted into the HindIII/SpeI restriction-digested pssAAV-2-hCMV-chI-EGFP-WPRE-bGHp(A) by three-way ligation.

**pssAAV-2-hSyn1-cdFLIIP(M20)-WPRE-hGHp(A) (p520).** The cdCitrine(M20) sequence including the C-terminal part of mgIB was isolated by BspEI/HindIII restriction digest from pssAAV-2-hCMV-chI-cdFLIIP(M20)-WPRE-bGHp(A) and inserted into BspEI/HindIII restriction-digested pssAAV2-hSyn1-FLIIP-WPRE-hGHp(A).

**pssAAV-2-hSyn1-ATeam1.03-WPRE-hGHp(A) (p252).** The N-terminal part of ATeam1.03 was isolated by NcoI/BsrGI restriction digest from pssAAV-2-hGFAP-hHBbI/E-ATeam1.03-WPRE-bGHp(A) (kind gift of Johannes Hirrlinger) and inserted into BsrGI/NcoI restriction-digested pssAAV2-hSyn1-ATeam1.03YEMK-WPRE-hGHp(A) (kind gift of Rodrigo Munje).

**pssAAV-2-hSyn1-chI-cdATeam1.03(M15)-WPRE-bGHp(A) (p567).** A PCR product of the synthetic cdcp173tdVenus(M15) fragment was generated with insertion of the EcoRI and AscI restriction sites at the N- and C-terminus, respectively, and the N-terminal part of ATeam1.03 was isolated by NcoI/BsrGI restriction digest from pssAAV-2-hGFAP-hHBbI/E-ATeam1.03-WPRE-bGHp(A) (kind gift of Johannes Hirrlinger). The two sequences were then inserted into AscI/NcoI restriction-digested pssAAV2-hSyn1-chI-GreenGlifon600-WPRE-bGHp(A) by three-way ligation.

**pssAAV-2-hSyn1-chI-hERT2-Cre-hERT2-WPRE-SV40p(A) (p325).** The hERT2-Cre(N-term) sequence was amplified by PCR, using Addgene #13777 as template, introducing a NheI site N-terminally, allowing to produce a NheI/BstBI fragment. A BstBI/NotIblunt fragment containing Cre(C-term)-hERT2 was isolated from Addgene #13777, and both fragments were cloned into EcoRV/NheI restriction-digested pssAAV-2-hSyn1-chI-EBFP2-WPRE-SV40p(A).

**pssAAV-2-hSyn1-chI-hERT2-iCre-cdhERT2-WPRE-bGHp(A) (p549).** An hERT2 fragment was isolated by NheI/SalI restriction digest from the p325 construct. The iCre-cdhERT2 sequence was isolated by SalI/HindIII restriction digest from a synthesized fragment (Supplementary Note 1). Both fragments were cloned into HindIII/NheI restriction-digested pssAAV-2-hSyn1-chI-MCS-WPRE-bGHp(A).

### Production, purification, and quantification of single-stranded (ss) AAV vectors
Single-stranded (ss) AAV vectors were produced and purified as previously described[65,66]. Briefly, human embryonic kidney (HEK) 293 cells[67] expressing the simian virus (SV) large T-antigen[68] (293 T) were transfected by polyethylenimine (PEI)-mediated cotransfection of AAV vector plasmids (providing the to-be-packaged AAV vector genome), AAV helper plasmids (providing the AAV serotype 2 rep proteins and the cap proteins of the AAV serotype of interest) and adenovirus (AV) helper plasmid pBS-E2A-VA-E4[69] (providing the AV helper functions) in a 1:1:1 molar ratio.

At 120 to 168 h post-transfection, HEK 293 T cells were collected and separated from their supernatant by low-speed centrifugation. AAV vectors released into the supernatant were PEG-precipitated one to two days at 4 °C by adding a solution of polyethylenglycol 8000 (8% v/v in 0.5 M NaCl) and completed by low-speed centrifugation (1 h at 3500 g/4 °C). Cleared supernatant was discarded and the pelleted AAV vectors were resuspended in AAV resuspension buffer (150 mM NaCl, 50 mM Tris-HCl, pH 8.5). HEK 293 T cells were resuspended in AAV resuspension buffer and lysed by Bertin's Precellys Evolution homogenizer in combination with 7 ml soft tissue homogenizing CK14 tubes (Bertin). The crude cell lysate was DENARASE (c-LEcta GmbH) treated (150 U/ml, 90–120 min at 37 °C) and cleared by centrifugation (10 min at 17.000 g/4 °C). The PEG-precipitated AAV vectors were combined with the cleared cell lysate and subjected to discontinuous density iodixanol (OptiPrep™, Axis-Shield) gradient (isopycnic) ultracentrifugation (2 h 15 min at 365'929 g/15 °C). Subsequently, the iodixanol was removed from the AAV vector containing fraction by 3 rounds of diafiltration using Vivaspin 20 ultrafiltration devices (100'000 MWCO, PES membrane, Sartorius) and 1x phosphate buffered saline (PBS), pH 7.4, supplemented with 1 mM MgCl$_2$ and 2.5 mM KCl (1x PBS-MK) according to the manufacturer's instructions. The AAV vectors were aliquoted and stored at -80 °C.

Encapsidated viral vector genomes (vg) were quantified using the Qubit™ 3.0 fluorometer in combination with the Qubit™ dsDNA HS Assay Kit (both Life Technologies). Briefly, 5 µl of undiluted (or 1:10 diluted in 1x PBS-MK) AAV vectors were prepared. Untreated and heat-denatured (5 min at 95 °C) samples were quantified according to the manufacturer's instructions. Intraviral (encapsidated) vector genome concentrations (vg/ml) were calculated by subtracting the extraviral (non-encapsidated; untreated sample) from the total intra- and extraviral (encapsidated and non-encapsidated; heat-denatured sample). The following physical titers were

measured: v520-6 [ssAAV-6/2-hSyn1-cdFLIIP(M20)-WPRE-hGHp(A)]: $6.9 \times 10^{12}$ vg/ml; v567-6 [ssAAV-6/2-hSyn1-chI-cdATeam1.03(M15)-WPRE-bGHp(A)]: $6.1 \times 10^{12}$ vg/ml; v252-9 [ssAAV-9/2-hSyn1-ATeam1.03-WPRE-hGHp(A)]: $6.2 \times 10^{12}$ vg/ml; vJS21-6 [ssAAV-6/2-hSyn1-FLIIP-WPRE-hGHp(A)]: $8.9 \times 10^{12}$ vg/ml; vJS20-6 [ssAAV-6/2-hSyn1-Laconic-WPRE-hGHp(A)]: $2.5 \times 10^{11}$ vg/ml; vLUR7-6 [ssAAV-6/2-hCMV-chI-cdATeam1.03tdVenus(M15)-WPRE-bGHp(A)]: $1.4 \times 10^{13}$ vg/ml; v520-PHP.eB [ssAAV-PHP.eB/2-hSyn1-cdFLIIP(M20)-WPRE-hGHp(A)]: $1.2 \times 10^{13}$ vg/ml; v325-8 [ssAAV-8/2-hSyn1-chI-hERT2-Cre-hERT2-WPRE-SV40p(A)]: $4.0 \times 10^{12}$ vg/ml; v549-retro [ssAAV-retro/2-hSyn1-chI-hERT2-iCre-cdhERT2-WPRE-bGHp(A)]: $7.9 \times 10^{12}$ vg/ml.

Identity of encapsidated genomes were verified and confirmed by Sanger DNA sequencing of amplicons produced from genomic AAV vector DNA templates (identity check).

### Next-generation sequencing and analysis

In order to analyze the abundance of the individual fluorescent protein nucleotide stretches within the recombined viral vector genomes, we conducted next generation sequencing. Original AAV samples were denatured by heat at 95 °C for 5 min, followed by gradual cooling to 30 °C for annealing of the single-stranded genomes to double-stranded DNA. In parallel, 10 ng of AAV plasmid DNA was utilized for PCR amplification. The amplification was carried out by a Phusion High-Fidelity DNA polymerase (Thermo Fisher Scientific) using the following oligonucleotides: forward primer (P17, 5′-ACTCAGCGCTGCCTCAG-3′), reverse primer (P174, 5′-TGTCAGTGCCCAACAGC-3′). The denatured AAV samples, as well as the PCR amplicons, were resolved on a 1% agarose gel. The pronounced recombined DNA bands (1271 bps, pJS21; 2752 bps, vJS21-6; 2517 bps, v252-9), as well as the non-recombined vJS21-6 band (4363 bps), were extracted with the help of the NucleoSpin Gel and PCR Clean-up kit (Macherey-Nagel) according to the manufacturer's instructions. 150-200 fmol of DNA per sample were sent to Microsynth AG, Switzerland, for the next generation sequencing. Briefly, a DNA library was prepared with the ligation sequencing kit V14 (SQK-LSK114) as per the manufacturer's instructions and the barcoded library was sequenced on Nanopore PromethION (PromethION Flow Cell R10.4.1, Oxford Nanopore Technologies). Basecalling, demultiplexing, and trimming of adaptor residuals were performed using DORADO, integrated into the MinKNOW software. Further post-basecalling analysis was performed with workflows available in the EPI2ME platform (https://github.com/epi2me-labs). Each sample provided more than 231,000 reads of high quality.

Recombination analysis was performed using a custom-made Python script. In short, we selected first the long reads that contained expected stretches downstream of the P17 binding site (3′ end of the human synapsin1 promoter) and upstream of the P174 binding site (5′ end of the WPRE element). For each individual read, the sequence between the selected stretches was extracted. All identical sequences were counted and grouped together. Within each group, we analyzed the nucleotides at positions in which the donor and acceptor FP sequences differ. Positions containing the nucleotide that corresponds to CFP or YFP were assigned the letter "C" or "Y", respectively, whereas in rare cases where the nucleotide did not match any of the two FPs (interpreted as a sequencing error), the whole group was discarded. Finally, by performing a weighted average of the number of C vs V at a given position, it was possible to identify the percentage of sequences that had recombined before that given position. The same analysis was conducted on a non-recombined construct to evaluate the percent threshold below which the occurrence of a sequence could be due to sequencing errors.

### Southern blot

AAV DNA was isolated by thermal disintegration of viral particles, followed by an annealing step to yield double-stranded DNA as follows: in a thermal cycler with lid heating (105 °C), 50 µl of viral preparation (appr. $5 \times 10^{12}$ viral particles per ml) were run through a thermal step gradient of 100 °C, 85 °C, 80 °C, 75 °C, 70 °C and 65 °C in 5 min steps. Subsequently, the DNA was purified with a PCR purification kit (Eurogentec, Liege, Belgium) and

afterwards subjected to restriction digest, using the combinations NheI/HindIII for FLIIP constructs and NcoI/HindIII for ATeam1.03 constructs. One ng of respectively digested DNA per lane were subjected to agarose gel electrophoresis, followed by a depurination step of the gel in 0.25 N HCl for 5 min and subsequent alkaline capillary transfer to a Biodyne B membrane (Pall, Port Washington, USA) using 0.4 M NaOH for 3 h, essentially as described by Reed and Mann[70]. The membrane was briefly washed with 2 x SSC and detection of DNA fragments containing GFP-related sequences was afforded by hybridization with a biotin-labeled ECFP cDNA, stringency washes in 0.1 x SSC, 0.1% SDS for $2 \times 10$ min and subsequent colorimetric detection according to the instructions of the provider (Biotin DecaLabel DNA Labeling Kit and Biotion Chromogenic Detection Kit, Thermo Fisher Scientific, Waltham, USA). For documentation and quantitation, the blots were recorded with a conventional flat-bed scanner (Epson Perfection V33, Epson, Nagano, Japan) at high resolution (1600 dpi) and directly used for documentation and quantification with the Analyze/Gels routine of ImageJ V2.1.0/1.53c[71] without any further processing.

### Two-photon ratiometric imaging in cells

HEK293T cells (ATCC CRL-3216) were cultured on glass bottom dishes (Cellvis D35-14-1.5-N) treated with poly-D-Lysine to improve adherence and incubated (37 °C, 5% $CO_2$) in DMEM medium (ThermoFisher) supplemented with 5% FBS (ThermoFisher). When confluency reached 50-60%, the cells were transfected with plasmids for the expression of either FLIIP (pssAAV-2-hCMV-chI-FLIP-WPRE-SV40p(A)) or cdFLIIP (pssAAV-2-hCMV-chI-cdFLIIP(M20)-WPRE-bGHp(A)) using the JetPEI transfection kit (Polyplus transfection) according to manufacturer instructions and imaged about 48 h after transfection.

For two-photon ratiometric imaging, the cell dishes were placed on top of a temperature-controlled mini bath chamber (slice mini bath chamber I, Luigs & Neumann) to ensure temperature stability.

The cells were superfused with a constant flow of ACSF (112 mM NaCl, 3 mM KCl, 1.25 mM $CaCl_2$, 24 mM $NaHCO_3$, 1.25 mM $MgSO_4$, 10 mM HEPES, pH 7.4), using two arduino-controlled[72] piezoelectric double-diaphragm pumps (mp6-hyb, Bartels Mikrotechnik). The buffer was continuously aerated by a gas mixture of 95% $O_2$ / 5% $CO_2$ and kept at a constant temperature of 34 °C. A fluid distributor (DiscofixC, Braun) was used to enable a seamless transition between different buffers during the imaging procedure. For the imaging protocol, the following supplements were added: baseline (ACSF, 2 mM glucose, 2 mM lactate, 23 mM sucrose), zero glucose condition (ACSF, 2 mM mannose, 2 mM lactate, 23 mM sucrose), and high glucose condition (ACSF, 25 mM glucose, 2 mM lactate). A total of three experiments (28 and 22 cells for FLIIP and cdFLIIP, respectively) were performed for each sensor. For the ATP-monitoring experiments using cdATeamDA, three experiments were performed at baseline and after 30 min incubation with metabolic blockers (2-deoxyglucose 10 mM + sodium azide 10 mM).

### Animal preparation

All experimental procedures were approved by the Veterinary Office of the Canton of Zurich and done in accordance with its guidelines. All experiments were conducted in compliance with all relevant ethical regulations for animal use. Two female wild-type mice (C57BL/6 J; Charles River) of age 3–8 months (20–25 g bodyweight) were used for imaging of glucose levels and dynamics by two-photon laser scanning microscopy (2PLSM). Animals were housed in groups of five under an inverted 12 h–12 h light-dark cycle with food and water ad libitum and were given at least one week to acclimatize to their housing before experimentation.

For surgical procedures, anesthesia was induced with 4% isoflurane and then maintained at a 2% level (in a 30%/70% oxygen/air mix) and mice were then placed into a stereotactic frame, using a heating pad to avoid hypothermia. For all procedures, buprenorphine (2 mg/kg) was delivered subcutaneously half an hour before stopping the isoflurane supply.

For head post implantation, the skull was first exposed by removing the skin and a bonding agent (ONE COAT 7 UNIVERSAL; Coltene) was

applied and polymerized using blue light (Dental WOODPECKER® curing light LED-F). A custom-made aluminum head post was attached to the skull with light curable dental cement (IPS Empress Direct Effect; Ivoclar vivadent AG), creating a "cap". The open skin was glued to the cap and antibiotic cream (Fucidin® 20 mg/1 g; LEO) was applied to prevent infection. Two days after the head-post surgery, a $3.5 \times 3.5$ mm$^2$ craniotomy was performed using a dental drill (Bien-Air) and solutions containing viral vectors were injected into the primary somatosensory cortex at 300–350 μm under the surface of the brain using a custom-made microinjection pump. A $3 \times 3$ mm$^2$ square sapphire glass plate (POWATEC GmbH) was placed over the brain and glued to the skull with dental cement, according to previously published protocols[73]. Mice were allowed to recover for three weeks before imaging. The ATeam1.03-YEMK image of Fig. 1b was acquired using a transgenic mouse (C57BL/6J-Thy1.2-ATeam1.03[YEMK])[74].

## Two-photon microscopy and FLIM setup
The 2PLSM images were acquired with a custom-made two-photon microscope[75], operated using a custom-written software based on ScanImage (Version 3.8)[76] and LabView (National Instruments) and coupled to a tunable femtosecond-pulsed laser (Chameleon Discovery, 80 MHz repetition rate,~ 100 fs pulse length) and a 16x water immersion objective (Nikon N16XLWD-PF, 0.8 NA, 3 mm WD). For ratiometric detection, the emitted light was spectrally separated using a set of three dichroics (F73-825, F38-560, F38-506; AHF Analysentechnik) and focused (LA1050-A1 and AL5040-A2; Thorlabs) on two PMTs (H10770PA-40sel, Hamamatsu), equipped with filters for CFP (475/50, AHF Analysentechnik) and YFP (542/50, AHF Analysentechnik). For FLIM detection, the CFP detector was replaced with a hybrid photomultiplier detector (Picoquant PMA-40mod), and the single photon pulses processed by a custom-made VHDL algorithm implemented on a Xilinx ZYNQ UltraScale ZCU102 FPGA board (courtesy of Dorian Amiet and Prof. Paul Zbinden, OST Rapperswill) capable of TCSPC acquisition in Time-Tagged Time-Resolved mode. The raw data were transmitted to a PC via TCP/IP over 1 Gbit/s ethernet and processed by a custom-made C ++ library to generate the FLIM images. Triexponential decay fitting using global analysis algorithms was performed using the FLIMfit software library[77] and an IRF function measured using second harmonic generation from KH$_2$PO$_4$ crystals at 950 nm excitation. For full-frame or ROI-based ratiometric analysis, the mean intensity of the YFP channel was divided by the mean intensity of the CFP one.

## In vivo and in vitro two-photon imaging
For in vivo 2PLSM imaging, mice were kept under anesthesia with 1.5% isoflurane and the breathing rate was kept at 55–65 breaths per minute. The core temperature was monitored and maintained at $37 \pm 0.5$ °C using a rectal probe and a heating blanket (Harvard Apparatus). For cell variability analysis, several planes were collected in two mice expressing both cdFLIIP and FLIIP at varying depths between 15 and 120 μm, with an excitation wavelength of either 870 or 925 nm, and ROIs were drawn manually to identify single neurons (870 nm: FLIIP $n = 526$, cdFLIIP $n = 447$; 925 nm: FLIIP $n = 526$, cdFLIIP n = 478). The intensities collected in the CFP and YFP channels for each ROI were used for correlation plots.

For glucose response experiments, D-glucose (50% w/v, 120 mL) was injected via tail vein over 1 min. Six experiments in two mice were collected in total, in which the FLIM and FRET curves were acquired in the same experiments, and areas expressing cdFLIIP and FLIIP were imaged in an alternated way (20 s per area, time resolution 40 s) upon excitation at 870 nm. The ratio value was calculated as the mean value of the YFP image divided by the mean value of the CFP one (obtained as sum of all temporal bins of the FLIM image), and later normalized to the mean of the 10 min baseline.

For visualization purposes, all FRET images in Figs. 1, 3 were denoised using the Noise2Void2 algorithm[78,79] before division of the acceptor and donor images (training on same data to be denoised, 2 channels, 25 epochs, patch size 64, mirrored padding).

## FLIM imaging of the G-PTEN sensor
HEK 293 T cells (American Type Culture Collection) in passage number 12–20 were cultured in DMEM supplemented with 10% FBS, 1% l-glutamic acid and 1% penicillin–streptomycin at 37 °C in 5% CO$_2$, and applied with 1 μl AAV GPTEN ($2 \times 10^{13}$ viral genomes per ml) or 1 μl AAV CD GPTEN ($2 \times 10^{13}$ viral genomes per ml) under the CMV promoter[63]. TBB (Tocris) was administered with a concentration of 50 μM in 2 ml. FLIM images were acquired using a 2pFLIM microscope which based on a Galvo-Galvo scanning system (Thorlabs) and a 2pFLIM module (Florida Lifetime Imaging), equipped with a Time-Correlated Single Photon Counting board (Time Harp 260, Picoquant). The microscope was controlled and fluorescent intensity quantified via the FLIMage software. For excitation, we used a Ti:sapphire laser (Chameleon, Coherent) at a wavelength of 960 nm to simultaneously excite AchLightG and mCherry. Excitation power was adjusted using a pockel cell (Conoptics) to 1.0–2.0 mW at 920 nm. Emission was collected with a $16 \times 0.8$ NA objective (Nikon), divided with a 565-nm dichroic mirror (Chroma), with emission filters of 525/50 nm and 607/70 nm, detected with two Photo-Multiplier Tubes with low transfer time spread (H7422-40p, Hamamatsu). Images were collected by $128 \times 128$. Each image was acquired at 2 ms/line, averaged over 24 frames. Data analysis was performed as previously reported[63].

## Immunohistochemistry and image acquisition
Mice were anesthetized with pentobarbital and transcardially perfused with 20 mL of ice-cold PBS (pH 7.4, (10X Dulbecco's)-Powder, Axon Lab AG, Baden-Dättwil, Switzerland) followed by 60 mL of ice-cold 2% PFA (Paraformaldehyde Granular, Electron Microscopy Sciences, Hatfield, PA) in PBS using a flow rate of 20 mL/min. Brains were dissected, halved coronally, post-fixed in 4% PFA for 3 h at 4 °C, cryoprotected in 30% sucrose in PBS for ~ 2 days at 4 °C, and then stored at −80 °C until needed for cutting. Coronal sections (40 μm thickness) were prepared using a microtome (Hyrax KS 34) and then stored in antifreeze solution (50 mM sodium phosphate buffer pH 7.4, 1 M glucose, 35% ethylene glycol and 3.5 mM sodium azide) at −20 °C until needed. Free floating sections were washed with 0.05% Triton X-100 (Sigma-Aldrich, Buchs, Switzerland) in Tris buffer (50 mM, pH 7.4) and then transferred to 0.3% Triton X-100 in Tris buffer supplemented with 5% donkey serum (blocking solution) for 1 h at room temperature. Sections were then incubated overnight with primary antibodies (Chicken polyclonal anti-GFP, Aves Labs Cat# GFP-1010, RRID: AB_2307313; 1:1000) diluted in the blocking solution at 4 °C. Following primary antibody incubation, sections were washed with Tris buffer and then incubated in 0.05% Triton X-100 (Tris buffer) containing Alexa Fluor 488-conjugated secondary antibodies (Alexa Fluor 488 AffiniPure Donkey Anti-Chicken IgY (IgG) (H + L), Jackson ImmuonoResearch Cat. No. 703-545-155, RRID: AB_2340375, 1:700) for 45 min at room temperature. Sections were mounted on microscope slides (SuperFrost Plus, Thermo Scientific) in Dako Fluorescence Mounting Medium (Dako, Jena, Germany). Confocal images were acquired using a confocal laser scanning microscope (Zeiss LSM 800) equipped with a 10X objective (Plan-Apochromat, NA 0.45).

## Combinatorial codon diversification
Combinatorial codon diversification was performed using the algorithm previously reported by Tang and Chilkoti[32], with identical parameters for the calculation of interaction energies. The algorithm was modified by replacing the original table that associates each amino acid to a list of all the codons encoding for it with a new table including only the codons whose relative abundance is above a user defined threshold, based on the codon usage table for mice extracted from the HIVE database[80]. Subsequence similarity was analyzed using-custom MATLAB algorithms.

## Statistics and reproducibility
Statistical analyses were performed using GraphPad Prism (version 10.2). Comparisons between two groups were assessed using Welch's t-test to account for unequal variances. For comparisons involving more than two groups, one-way ANOVA followed by Tukey's multiple comparisons test

was used. Significance levels are indicated as follows: ****$p < 0.0001$; ***$p < 0.001$; **$p < 0.01$; *$p < 0.05$; ns, not significant.

Data from dynamic experiments are presented as mean ± SEM. All animal experiments were conducted using at least two mice as independent biological replicates. Cell-based experiments were performed with at least three independent biological replicates.

Single-cell analyses were based on hundreds of individual cells from two mice per dataset to capture cell-to-cell variability. Sex-based effects were not investigated, as they were deemed irrelevant. Sample sizes were considered adequate based on the observed large effect sizes. Detailed sample sizes and statistical annotations are provided in the corresponding figure legends. No experiments were excluded and no blinding was applied.

## Reporting summary

Further information on research design is available in the Nature Portfolio Reporting Summary linked to this article.

## Code availability

The code for the analysis of FRET and FLIM data is available at https://gitlab.com/einlabzurich/flimanalysis, while the code to perform codon diversification is available at http://chilkotilab.pratt.duke.edu/codon-scrambler. FLIM analysis for the GPTEN was performed using a custom C# software available at https://github.com/ryoheiyasuda/FLIMage_public. Versions of the code used in this work are available as a Zenodo repository (https://doi.org/10.5281/zenodo.16368455).

## Data availability

Data associated with the paper, including sequencing results, is available as a Zenodo repository (https://doi.org/10.5281/zenodo.16368455). For cases in which gels were cropped, the uncropped images are reported in FigureSI9.

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

## Acknowledgements

We thank Nicholas C. Tang and Prof. Ashutosh Chilkoti for sharing their code and helping with its usage. A.S.S. was supported by the Swiss National Science Foundation (SNSF; Eccellenza 187000). T. Laviv is supported by Israel Science Foundation (ISF) grants 1384/21 and 1385/21. All authors declare no conflicts of interest.

## Author contributions

J.D., L.V., M.R., and J.C.P. performed the cloning procedures and viral preparations, L.V., M.R., and M.A. performed gel electrophoresis and Southern blotting, A.E., and Z.J.L. performed in vivo experiments and helped identifying the recombination problem, H.Z. performed in vivo experiments for the AAV-PhP.eB viruses and immunohistochemical staining, P.I. performed in vitro experiments in HEK cells, A.E., R.M.M. and F.V.P. performed data analysis and visualization for in vivo experiments, J.D., L.V., M.R. and L.R. analyzed sequencing results, T.K. and T.L. performed experiments on the G-PTEN sensor, L.R. conceived the study, and performed coding and analysis of computational results, L.R., B.W., and A.S.S. supervised experiments and wrote the manuscript. All authors contributed to revising the manuscript.

## Competing interests

The authors declare no competing interests.
