## [Transparent Peer Review file · Communications Biology]

Abundance-Biased Codon Diversification prevents recombination in AAV production and ensures robust in vivo expression of functional FRET sensors

Corresponding Author: Dr Luca Ravotto

This manuscript has been previously submitted at another journal. This document only contains information relating to versions considered at Communications Biology.

Version 0:

Reviewer comments:

Reviewer #1

(Remarks to the Author)

Dernic et al. describe an important issue associated with the widely employed AAV in that sequences with high similarity within the AAV cargo can recombine. The authors show that this can be eliminated by diversifying the sequences. Regarding the sequence diversification, the authors use a published algorithm with small modifications and rename it "ABCD". Overall, the results are of potential importance. However, the manuscript can benefit from a number of improvements, as below:

Major.

1. The authors use existing codon diversify algorithm with minor modifications (constraining the possible choice of codons). The reviewer feels that the degree of improvement/modification does not warrant a renaming/rebranding the method. The use of existing algorithm should also be made transparent in the abstract.
2. ECFP is rarely used anymore in the field. Most sensors use mCerulean or mTurquoise, which bears additional mutations. Would these FPs recombine with mVenus?
3. Related to above, what are the f value for common FP pairs, such as mCerulean-mVenus, mTurquoise-mVenus, etc?
4. Since Laconic has a high f value and still exhibits no recombination, is it necessary to set the threshold to $f \sim 10^{10}$?
5. Related to above, what is the f value where recombination starts to happen? And how does it compare to the f value between an FP that is optimized using two independent online algorithms? Note that many companies offer codon optimization/diversification and the sequence from different websites are often different. Is this sufficient?
6. Fig. 1b: The variability needs to be quantified and normalized to the average ratio in order to compare across sensors. E.g., ratio variabilities between 0.1-0.4 and 0.5-2 are equal, but this will show very differently in the representative figures using the same set of color map scale.
7. Fig. 1c: Can the PCR be subjected to next generation sequencing so that the presence of sequence from both FPs indeed happen at the single DNA molecule level? This is versus mixed population due to mis-annealing during PCR. It will also allow for the analysis of the exact recombination point.
8. Line 99: the nature of the mutations should be described. For example, an analyte binding-disabled sensor would show more consistent ratio compared to the wildtype sensor.
9. Fig. 3e: although cdFLIIP appears to be less variable compared to FLIIP, but the data does not support the claim that changing wavelength increase variability for FLIIP (line 340).
10. Fig 3g and 4d: why for the cdFLIIP, the baseline is not stable? If extrapolating the baseline, the response really is not that large...

Other.

1. Lines 61-63, 67-68: without further justification, these sentences are not consistent with the common thoughts. Most would think that the rare use of FRET sensors is largely because of their smaller dynamic range and larger size compared to single

fluorophore sensors.

2. Fig. 1a: can there be a schematic showing where the PCR primers are located? This helps the interpretation of whether the result may be explained by mis-annealing of primers and the shorter sequences are preferentially amplified.
3. Fig. 1f: The product can be made larger for visualization. Currently the differences between products are hard to see.
4. Line 343: "FLIM is the most robust method to quantitatively estimate concentrations". It should be specified that it is analyte concentration (vs sensor concentration).
5. Line 462: "... of the Cre protein." Please include figure reference here to support the claim.
6. Line 471-472: the claim is too strong. E.g., would a 20 bp repeated sequence trigger recombination? What is the length threshold? This is not addressed in the manuscript.

Reviewer #2

(Remarks to the Author)

Dernic et. al. have provided a paper that outlines an algorithm for codon optimizing fluorescent FRET pairs proteins for AAV based expression. Recombination of AAV genome is a well-studied phenomena that arises when there are stretches of highly similar DNA sequences within the AAV genome. Many FRET pair proteins are derived from same protein scaffold making their sequences at both DNA and protein level similar. The authors used a previously published algorithm (published by Tang and Chilkoti) that utilizes codon redundancy to minimize the sequence similarity at the DNA level, hence reducing the percentage of AAV recombination.

Below are some issues that could be addressed to strengthen the manuscript.

1. The authors arrive at the $f \sim 10^{10}$ based on calculation from a single set of FRET pairs that they hypothesize doesn't undergo recombination. If this number is an important threshold number that is used to further build the algorithm and optimize the sequences for FRET pairs, it would be good to use few other FRET pairs to make sure this number is robust.
2. Authors should show the activity of lactate FRET sensor Laconic in Fig 1b, but referenced source doesn't perform radiometric two-photon microscopy images of neurons. As it stands it does not quantify difference in cell-to-cell variability and nuclear localization. It was hard to understand if this was done in transgenic mouse or HEK293 cells.
3. Aim of the figure 1f is clear, but the figure and captions need refinement and clarification. The DNA inside the AAV shouldn't be circular plasmid as shown as it represents incorrect structure of the DNA that gets packaged inside AAV.
4. Could authors clarify why they fixed the N-term sequence when they diversified the sequence as indicated in the manuscript?
5. Recombination is not just based on cargo, but ITR mediated recombination is also a commonly observed phenomenon. It would be great for the authors to comment on that as well.
6. There seems to be greater presence of long stretches along the diagonal on the cdCitrine (M20) dot plot compared to the Citrine dot plot indicating sharing of a common ancestor in Figure2f but I believe the authors are claiming the opposite. May be the authors could help clarify this?
7. In figure 3b, to paint a more complete relationship from the gel and southern blot, it would be helpful to include cdATeamDA in the sample blot as the ATeam and cdATeam blots.

Overall Comment.

While the platform itself is very promising and a clever use of Tang and Chilkoti's algorithm, performing the same optimizing with ATeam compared to cdATeam and cdATeamDA would further strengthen the author's claim that their refined algorithm works for codon diversification across different FRET-based biosensors susceptible to recombination. While showing in vivo FLIM imaging of AAV-delivered FLIIP supports the lack of recombination through the lack of variability in FLIM values and up-regulation in FLIM responses, the indication that this can work for other FRET-based biosensors targeting different pathways would be more revealing of the applicability of the algorithm.

Reviewer #3

(Remarks to the Author)

Dernic et al. introduce a novel strategy for the sequence design of genetically encoded Förster Resonance Energy Transfer (FRET) sensors for engineering functional adeno-associated virus (AAV) in vivo. This approach, designated ABCD, computationally designs diverse DNA sequences for multiple fluorescent proteins in the sensor, effectively mitigating recombination issues while maintaining codon optimization for specific target organisms. Employing these sensors, the authors successfully engineered AAVs and demonstrated their in vivo functionality through fluorescence intensity and lifetime measurements, potentially addressing a challenge in AAV-based in vivo imaging.

Overall, this manuscript offers a substantial contribution to our understanding of the importance of DNA sequence in developing functional AAVs for in vivo imaging. Its implications are noteworthy for Communicational Biology. However, I recommend addressing the following major and minor issues prior to publication:

Major issues:

1. Lines 89–, Fig. 1: The manuscript describes an examination of viral vectors via gel electrophoresis and in vivo expression, indicating loss of parts of the FRET sensor gene. It is

crucial for the authors to verify and present the original sequence of the FRET sensor gene in the AAV vector plasmid utilized for AAV preparation to confirm its integrity.

2. Lines 221–233: The discussion around the double-acceptor strategy should more thoroughly address the size constraints inherent in AAV engineering. Given the AAV's typical capacity (~4 kb), including essential elements like the promoter, WPRE, and polyA signal, the feasibility of incorporating three fluorescent proteins and an analyte binding domain into a FRET sensor via the ABCD method requires empirical validation through in vivo experiments with AAV-encoded sensors.

3. Figs. 3g and 4d: The cdFLIIP signal profiles in response to glucose injection raise questions about its functional efficacy as a glucose sensor in vivo. The similarity in signal slopes pre- and post-injection does not convincingly support the in vivo functionality of the FRET sensor derived from the ABCD method. A critical discussion and appropriate manuscript revision are needed to address this.

4. Lines 584–586: The citation of previous works (refs. 66 and 67) as demonstrations of the ABCD approach appears misleading, as these references do not explicitly mention this method in the context of FRET sensor design. A revision of this statement is necessary for accuracy.

Minor issues:

1. Ensure consistent use of full terms and abbreviations throughout the manuscript (e.g., fluorescent proteins (FPs), genetically encoded sensors (GESs), calcium (Ca²⁺) etc.).

2. The sequence of figure panels in the text appears disorganized, leading to reader confusion. Aligning the figure references in the text with their corresponding panels in a logical sequence is advised.

3. Line 50: "The first FRET sensor" should be more accurately described as "The first FRET GES" to acknowledge the prior existence of chemical dye-based FRET sensors.

4. Line 63: Amend "GCamp" to "GCaMP."

5. Fig. 4d: Reorient the Y-axis of the graph to display values from lower (smaller τ) to upper (larger τ) for clarity.

6. Lines 505–506: It is generally understood that the analyte binding domain and linkers in a FRET sensor primarily influence the apparent dissociation constant (K_d), rather than the FP domain. If the FP domain does significantly affect K_d , please provide relevant citations for support.

Version 1:

Reviewer comments:

Reviewer #1

(Remarks to the Author)

In this revised manuscript, Deric et al. have performed additional experiments, new analysis and textual edits that have addressed my prior comments. I support the publication of this manuscript. However, there are still a couple of minor points that I would like them to be addressed.

1) The FLIIP image in Fig. 1b is a duplication of that in Fig. 3d. Can one of them be replaced?

2) "glc iv" in Figure 3g is probably better changed to "iv" since saline injection is also used.

3) The n for some experiments is missing. E.g., in legend of Fig. 4d, please provide number of neurons/mice.

Reviewer #3

(Remarks to the Author)

The authors have satisfactorily addressed all of the concerns raised in the previous review. The revised manuscript is now suitable for publication in its current form.

Reviewer #4

(Remarks to the Author)

I had the chance to look at the manuscript from Deric et al only as a revised manuscript, and not at the time of the first submission.

The new Figure 3 and the Tables, together with additions to the narrative, greatly improve the quality of the paper. The proposed strategy is clever and holds promises for biosensor optimization on a wider scale.

There are a few points that remain to be solved:

1. Despite the answer provided by the authors, I also believe that there are long stretches along the diagonal of ECFP/cdCitrine (M20) plot in Fig. 2f although there is no red line highlighting them. Can the authors comment on this? The diagonal pattern is evident by eye.

2. I appreciate the reference to the recently published Nat Methods paper, which shows a PTEN biosensor built by taking advantage of the strategy implemented by the authors. However, this does not fully answer the issue of how generalized this strategy can be. The Nat Methods paper reports on a novel sensor, and unless I'm mistaken, there is no comparison between an unmodified and a cd-modified version of the sensor. Concerning ATeam, I acknowledge that the only

fluorescent product upon recombination is the fluorescent one. However, can codon diversification further stabilize the readout of the sensor and make it more robust in cells where expression levels of the sensor are naturally low, for instance? This would substantiate that "losing" a fraction of the unmodified ATeam would be detrimental to the S/N ratio and therefore, to correctly estimate FRET. In this case, expressing cdATeam would be a clear advantage to improve FRET detection. Overall, I believe that a second example highlighting the breadth of the codon optimization strategy would be beneficial to generalize the strategy.

3. Fig. 3g and 4d. Why is the FLIIP–saline condition missing? Do we see a difference between glucose and saline with the original sensor? If that were the case, is the delta of the response between saline and glucose maintained when FLIIP and cdFLIIP are compared? I believe this control condition is necessary to fully appreciate the improved response of the cdFLIIP construct.

Version 2:

Reviewer comments:

Reviewer #4

(Remarks to the Author)

I thank the Authors for taking the time to provide extensive answers to my questions, and for introducing critical data in Supplementary Fig. 8. This is a very nice story, congratulations!

Reviewer 1

Reviewer #1 (Remarks to the Author):

Dernic et al. describe an important issue associated with the widely employed AAV in that sequences with high similarity within the AAV cargo can recombine. The authors show that this can be eliminated by diversifying the sequences. Regarding the sequence diversification, the authors use a published algorithm with small modifications and rename it "ABCD". Overall, the results are of potential importance. However, the manuscript can benefit from a number of improvements, as below:

Major.

1. The authors use existing codon diversify algorithm with minor modifications (constraining the possible choice of codons). The reviewer feels that the degree of improvement/modification does not warrant a renaming/rebranding the method. The use of existing algorithm should also be made transparent in the abstract.

We agree with the reviewer that the algorithm used is a constrained version of Tang and Chilkoti's method. The original work was already highlighted in the text and by explicitly naming the paper in Figure 2a. On the same line, we modified the abstract stating that our algorithm is "a modification of a previously reported codon diversification" one. However, we respectfully disagree with the fact that the algorithm does not need a renaming, as the ability to control the abundance bias effectively puts the algorithm "in between" codon diversification and codon optimization, with the importance of one vs the other tuned by the bias value. In particular, the concept of ABCD goes beyond the specific algorithms used, but rather refers to the general concept of applying a codon optimization bias to a codon diversification algorithm.

2. ECFP is rarely used anymore in the field. Most sensors use mCerulean or mTurquoise, which bears additional mutations. Would these FPs recombine with mVenus?

Indeed, more advanced versions of FPs carry more mutations with respect to the original avGFP sequence, and thus might in principle be less likely to recombine. Nevertheless, they still carry long stretches of identical sequences. To quantitatively investigate whether newer FPs could provide a solution to recombination, we ran our analysis for different combinations of more modern donors and acceptors. In all cases, very long identical sequences were still present, and the algorithm returned an f value $> 10^{308}$, an indication of very high similarity.

The following paragraph was added to the text:

In principle, replacing the donor or acceptor FPs with more modern FP variants could reduce recombination, as newer variants carry additional mutations that reduce similarity. To investigate this idea, we have run similarity analysis for all combinations between the cyan variants mTurquoise2, Aquamarine and mCerulean3 and the yellow variants Citrine, YPet and SYFP2. For pairs involving YPet, the longest identical stretch was 72 bp, compared to all other cases in which stretches of more than 220 bp were identified. Nevertheless, in all cases the f values were $> 10^{308}$, indicating that using more modern FPs is not a viable strategy to reduce recombination.

3. Related to above, what are the f value for common FP pairs, such as mCerulean-mVenus, mTurquoise-mVenus, etc?

This point was addressed together with the previous one.

4. Since Laconic has a high f value and still exhibits no recombination, is it necessary to set the threshold to $f \sim 10^{10}$?

Based on the current data, it is impossible to say whether the value of 10^{10} obtained for Laconic is a hard limit. In their original work on PCR amplification, Tang and Chilkoti tested a large number of sequences and observed a sharp decline of recombination below 10^7 . In our study, we have a more polarized situation, in which either the f-value is incredibly high, or it is below the limit reported by Tang and Chilkoti. In our opinion, we envision the use of the f-value not as an absolute number that would guarantee no recombination, but rather as a guidance to compare the level of diversity with some reference values (the “natural diversity” of Laconic and the “PCR threshold”), after which experimental testing is anyways required to confirm the elimination of recombination. Nevertheless, to further investigate the concept of “natural diversity”, we conducted a similarity analysis on pairs of FPs originating from different organisms, and modified the corresponding paragraph to:

An f value analysis of pairs of FPs coming from different organisms (Table S11) shows values ranging from $\sim 10^{13}$ to $\sim 10^5$, with mNeonGreen and mStayGold appearing particularly diverse from other families of FPs, and among themselves. Thus, while higher f values might be sufficient to avoid recombination (as shown in Laconic), as a conservative estimate, we consider the value of $\sim 10^7$ identified by Tang and Chilkoti in their PCR study a suitable target.

5. Related to above, what is the f value where recombination starts to happen? And how does it compare to the f value between an FP that is optimized using two independent online algorithms? Note that many companies offer codon optimization/diversification and the sequence from different websites are often different. Is this sufficient?

The first part of the question was answered in the point above. Regarding the usage of other algorithms, we believe that this is certainly a possibility. Nevertheless, to the best of our knowledge, we found only one company that was able to offer a combination of diversification and optimization. In addition, no information was provided as to how the algorithm works. In our view, there is always a chance that any algorithm that modifies the DNA sequence of a protein would solve the recombination issue. On the other hand, the one presented by Tang and Chilkoti offers several advantages, as it was designed specifically for avoiding recombination focusing not only on sequence identity but on interaction energy (a feature that reduces at the same time the risk of formation of secondary RNA structures), while at the same time allowing to easily modify the available codon choice (in principle, codons could be excluded with more sophisticated rules than our abundance bias) and to exclude unwanted sequences.

6. Fig. 1b: The variability needs to be quantified and normalized to the average ratio in order to compare across sensors. E.g., ratio variabilities between 0.1-0.4 and 0.5-2 are equal, but this will show very differently in the representative figures using the same set of color map scale.

We apologize for this lack of precise quantitation. The aim of Figure 1b was not to provide a quantitative comparison but rather to explain to the reader the observations that triggered our interest in delving deeper into the problem. As both color scales are between 0 and 2, we believe that the visual observation justifies our qualitative claim. Nevertheless, in this case a different variability could also in principle originate from the different variability in ATP and glucose concentrations. To avoid any confusion, we eliminated any reference to variability when mentioning

the observation in Figure 1b, while a more detailed analysis of variability was performed for the data presented in Figure 3, in which the differences originate from the elimination of recombination.

7. Fig. 1c: Can the PCR be subjected to next generation sequencing so that the presence of sequence from both FPs indeed happen at the single DNA molecule level? This is versus mixed population due to mis-annealing during PCR. It will also allow for the analysis of the exact recombination point.

We thank the reviewer for prompting us to perform additional sequencing. NGS sequencing was performed on PCR products of AAV preparations and non-recombined plasmids, as well as on thermally denatured AAV samples (no amplification steps). The results are included in Figure SI4.

We discovered that recombination indeed occurs both during AAV preparation and PCR, a fact we previously overlooked. Nevertheless, we confirmed the presence of individual sequences combining CFP and YFP stretches. A statistical analysis of those sequences allowed us to identify the percentage of recombination as function of the position on the sequence. The strong linear correlation between position and recombination percentage suggests that there is no particular preference for the position in which recombination occurs, at least for our constructs.

The following paragraph was added to the paper:

Since the plasmids used for the viral preparations were found to contain the correct sequences (Figure SI3), recombination could have occurred during PCR amplification and/or AAV preparation. Indeed, PCR amplification of a non-recombined plasmid containing the FLIIP sequence leads to the presence of recombined products following gel electrophoresis (Figure SI4a). To assess the possibility of recombination occurring also during AAV production, original AAV samples (without amplification) were analyzed by gel electrophoresis. For both FLIIP (Figure SI4a) and ATeam (Figure SI4e), recombination bands were detected. By performing next-generation sequencing on the recombined constructs, we confirmed the presence of individual sequences combining CFP and YFP subsequences (Figure SI4c, f). Furthermore, our data show that the recombination probability grows linearly as a function of the nucleotide position (Figure SI4d, g), suggesting that, at least in our experimental set-up, recombination occurs with approximately equal frequency at any position.

8. Line 99: the nature of the mutations should be described. For example, an analyte binding-disabled sensor would show more consistent ratio compared to the wildtype sensor.

We specified that the mutations increase the affinity for ATP. We agree that this could also impact variability, and we eliminated any reference to variability in this context (see point 6).

Here is the new text:

This behavior is in stark contrast with what we previously observed with the lactate FRET sensor Laconic and intriguingly with the ATP sensor ATeam1.03YEMK (Figure 1b, right), which only differs from the variant used in this study only by a few mutations in the binding pocket that increase the affinity for ATP.

9. Fig. 3e: although cdFLIIP appears to be less variable compared to FLIIP, but the data does not support the claim that changing wavelength increase variability for FLIIP (line 340).

We apologize for the lack of clarity. We have computed statistics on the distributions of the ratio values for all four conditions (Figure 3e). In particular, we have calculated the coefficient of variation, as ratio between SD and mean, proving that for FLIIP there is an increase in the CV from

0.22 to 0.31 when passing from 870 to 925 nm excitation. On the contrary, for cdFLIIP both wavelengths show a coefficient of variation of about 0.06, which is both much smaller than the one of FLIIP and independent from the excitation wavelength.

10. Fig 3g and 4d: why for the cdFLIIP, the baseline is not stable? If extrapolating the baseline, the response really is not that large...

We agree with the reviewer that the baseline is not stable. This was already observed in ref.67, and its origin is unclear. To prove that despite this drift the sensor responds to glucose, we performed additional control experiments injecting saline using an identical bolus i.v. injection protocol. We updated the relevant figures and added the following explanation in the text:

Upon glucose injection, cdFLIIP shows ratiometric and lifetime changes compatible with the expected increase in glucose concentration. However, as the curve before injection is not perfectly stable, we performed a control injection using saline. Indeed, a drift of the signal is observed, but the larger change upon glucose injection confirms the sensor's functionality.

An important point here is that we do not want to claim that FLIIP is necessarily the best sensor for glucose in terms of sensitivity or accuracy (which would need to be evaluated in a separate study). The goal of our experiments is rather to demonstrate that the elimination of recombination has restored the functionality of FLIIP as a glucose sensor (and by extension, any sensor with a similar design that would suffer from recombination).

Other.

1. Lines 61-63, 67-68: without further justification, these sentences are not consistent with the common thoughts. Most would think that the rare use of FRET sensors is largely because of their smaller dynamic range and larger size compared to single fluorophore sensors.

While it is true that FRET sensors have a larger size, it is generally possible to include them in AAV sensors, as shown in this and other studies, so this problem is limited to applications that require very large promoters or additional genes in the AAV cargo. As for the dynamic range, while the response of FRET sensors is generally numerically lower than that of single-FP sensors, it is also more robust to in-vivo imaging artifacts, and thus they might compare favorably with single-FP sensors even when with a smaller dynamic range. A response of 50% $\Delta R/R$ would be considered low for single-FP sensors (in this case $\Delta F/F$), but it is often sufficient for a FRET sensor. Nevertheless, we agree that our sentence is somewhat arbitrary. We replaced "explaining the lack of adoption" with "hindering a more widespread adoption".

2. Fig. 1a: can there be a schematic showing where the PCR primers are located? This helps the interpretation of whether the result may be explained by mis-annealing of primers and the shorter sequences are preferentially amplified.

This data is reported in Supplementary Figures 1 and 2. To make this clearer, we added a reference to those figures in the caption of Figure 1a.

3. Fig. 1f: The product can be made larger for visualization. Currently the differences between products are hard to see.

We have updated the images to improve visualization and correct the fact that we mistakenly added a plasmid in the viral vectors, instead of a linear DNA sequence.

4. Line 343: “FLIM is the most robust method to quantitatively estimate concentrations”. It should be specified that it is analyte concentration (vs sensor concentration).

We added the term “analyte” to the sentence.

5. Line 462: “... of the Cre protein.” Please include figure reference here to support the claim.

We added a reference to Figure S17.

6. Line 471-472: the claim is too strong. E.g., would a 20 bp repeated sequence trigger recombination? What is the length threshold? This is not addressed in the manuscript.

At this stage, it is impossible to determine whether an exact threshold length exists. For this reason, we used the wording “can occur” rather than “occurs”. Nevertheless, the argument that short identical sequences will likely not recombine is valid, and we replaced “sequences” with “long sequences” to make this more clear.

Reviewer 2

Reviewer #2 (Remarks to the Author):

Dernic et. al. have provided a paper that outlines an algorithm for codon optimizing fluorescent FRET pairs proteins for AAV based expression. Recombination of AAV genome is a well-studied phenomena that arises when there are stretches of highly similar DNA sequences within the AAV genome. Many FRET pair proteins are derived from same protein scaffold making their sequences at both DNA and protein level similar. The authors used a previously published algorithm (published by Tang and Chilkoti) that utilizes codon redundancy to minimize the sequence similarity at the DNA level, hence reducing the percentage of AAV recombination.

Below are some issues that could be addressed to strengthen the manuscript.

1. The authors arrive at the $f \sim 10^{10}$ based on calculation from a single set of FRET pairs that they hypothesize doesn't undergo recombination. If this number is an important threshold number that is used to further build the algorithm and optimize the sequences for FRET pairs, it would be good to use few other FRET pairs to make sure this number is robust.

This point was also raised by Reviewer 1. We report here the same answer.

Based on the current data, it is impossible to say whether the value of 10^{10} obtained for Laconic is a hard limit. In their original work on PCR amplification, Tang and Chilkoti tested a large number of sequences and observed a sharp decline of recombination below 10^7 . In our study, we have a more polarized situation, in which either the f -value is incredibly high, or it is below the limit reported by Tang and Chilkoti. In our opinion, we envision the use of the f -value not as an absolute number that would guarantee no recombination, but rather as a guidance to compare the level of diversity with some reference values (the "natural diversity" of Laconic and the "PCR threshold"), after which experimental testing is anyways required to confirm the elimination of recombination. Nevertheless, to further investigate the concept of "natural diversity", we conducted a similarity analysis on pairs of FPs originating from different organisms, and modified the corresponding paragraph to:

*An f value analysis of pairs of FPs coming from different organisms (Table S11) shows values ranging from $\sim 10^{13}$ to $\sim 10^5$, with *mNeonGreen* and *mStayGold* appearing particularly diverse from other families of FPs, and among themselves. Thus, while higher f values might be sufficient to avoid recombination (as shown in Laconic), as a conservative estimate, we consider the value of $\sim 10^7$ identified by Tang and Chilkoti in their PCR study a suitable target.*

2. Authors should show the activity of lactate FRET sensor Laconic in Fig 1b, but referenced source doesn't perform radiometric two-photon microscopy images of neurons. As it stands it does not quantify difference in cell-to-cell variability and nuclear localization. It was hard to understand if this was done in transgenic mouse or HEK293 cells.

We apologize for the confusion. Figure 1b depicts the ATP sensor ATeam in transgenic mice. We indicated this in the caption but forgot to mention it in the methods. We amended this. The point on variability was also raised by Reviewer 1. The aim of Figure 1b was not to provide a quantitative comparison, but to explain to the reader which was the visual observation that triggered the present study. We eliminated all references to variability from the text when referring to Figure 1. To address the criticism, a more quantitative analysis of variability and nuclear localization was performed for Figure 3, to quantify the difference between the recombined and non-recombined

FLIIP sensor. We also included arrows in Figure 1 and 3 to visually highlight the presence of darker circles in the somata of neurons, a clear morphological feature of the nucleus.

The following paragraph, together with the changes to Figure 3 and its caption, reflect the changes:

Recombination is clearly observable in the higher cell-to-cell variability (Figure 3d,e) and nuclear localization (Figure 3d,f) in FLIIP. Both effects are due to the presence of a mixture of FPs, which confer the variability and are small enough to cross the nuclear membrane. For cdFLIIP, switching the wavelength from 870 to 925 nm caused a change in average ratio but not an increased variability (Figure 3e). On the other hand, the significant increase in variability observed for FLIIP is proof of the presence of multiple emissive species, featuring different excitation efficiency at different wavelengths. The lack of nuclear localization is clearly visible in the images as the dark round spot in the somata, a well-known morphological feature. To give a more quantitative estimate, we calculated the percentage of pixels within each soma whose intensity is below a certain fraction of the mean intensity of the same area, showing a clear difference between the two sensors (Figure 3f).

3. Aim of the figure 1f is clear, but the figure and captions need refinement and clarification. The DNA inside the AAV shouldn't be circular plasmid as shown as it represents incorrect structure of the DNA that gets packaged inside AAV.

We thank the reviewer for spotting this mistake. We have updated the images to correct it and improve visualization.

4. Could authors clarify why they fixed the N-term sequence when they diversified the sequence as indicated in the manuscript?

The reason is that the codon diversification code by Tang and Chilkoti works by taking a sequence of amino acids and generating a DNA sequence that maximize diversity within itself (as it was originally conceived for repetitive AA sequences). In our case, we needed not only that, but also the maximization of diversity with respect to the "original" DNA sequence of FPs derived from avGFP. To achieve that, we exploited a feature of the code that allowed to "freeze" a certain DNA sequence at the N-terminus of the AA sequence to be diversified. While this "frozen" sequence is not changed during optimization, it is used to calculate the f-value at each step, thus ensuring that the generated sequence is maximally diverse also with respect to the frozen N-terminal one. An alternative approach would have been to just use the AA sequence of a CFP-YFP construct, thus diversifying both FPs at the same time. This approach however was way too costly computationally and failed due to memory limitations.

5. Recombination is not just based on cargo, but ITR mediated recombination is also a commonly observed phenomenon. It would be great for the authors to comment on that as well.

ITR mediated recombination is indeed a common feature of viral biology and part of the normal process of AAV preparation. We prefer not to discuss it in the manuscript to avoid creating confusion, as the phenomenon we describe is happening within the cargo and should be considered a problem to avoid.

6. There seems to be greater presence of long stretches along the diagonal on the cdCitrine (M20) dot plot compared to the Citrine dot plot indicating sharing of a common ancestor in Figure2f but I believe the authors are claiming the opposite. May be the authors could help clarify this?

We believe that Figure 2f is correct. The plot on the left, showing the Citrine-ECFP pair, displays the long identical stretches, while the one on the right, displaying the codon-diversified cdCitrine-ECFP pair, doesn't.

7. In figure 3b, to paint a more complete relationship from the gel and southern blot, it would be helpful to include cdATeamDA in the sample blot as the ATeam and cdATeam blots.

In the case of cdATeamDA, we did not perform Southern blot as only a single band was observed when thermally denatured viral samples were run through gel electrophoresis. We added a comparison of the results of this experiment between ATeam, cdATeam and cdATeamDA in Figure S15. As part of the manuscript reorganization, with the addition of the requested quantitative analyses of variability and recombination to Figure 3, the Southern Blots have also been moved to Figure S15.

Overall Comment.

While the platform itself is very promising and a clever use of Tang and Chilkoti's algorithm, performing the same optimizing with ATeam compared to cdATeam and cdATeamDA would further strengthen the author's claim that their refined algorithm works for codon diversification across different FRET-based biosensors susceptible to recombination. While showing in vivo FLIM imaging of AAV-delivered FLIIP supports the lack of recombination through the lack of variability in FLIM values and up-regulation in FLIM responses, the indication that this can work for other FRET-based biosensors targeting different pathways would be more revealing of the applicability of the algorithm.

We thank the reviewer for his/her positive comment. In the paper, we have demonstrated at the sequencing and gel electrophoresis level that recombination occurs and can be eliminated in FLIIP, ATeam and an hERT2-Cre-hERT2 system. In the case of ATeam, we did not perform any photophysical comparison as we know that even the recombined one works well, as the only fluorescent product in the recombined batch is the fully functional sensor. At the time of submission, we did not have any other sensor example, but a paper utilizing our method to codon-diversify a sensor for PTEN has been recently published in Nature Methods (<https://www.nature.com/articles/s41592-025-02610-9>). A reference to this work was added in the conclusions.

Reviewer 3

Reviewer #3 (Remarks to the Author):

Dernic et al. introduce a novel strategy for the sequence design of genetically encoded Förster Resonance Energy Transfer (FRET) sensors for engineering functional adeno-associated virus (AAV) *in vivo*. This approach, designated ABCD, computationally designs diverse DNA sequences for multiple fluorescent proteins in the sensor, effectively mitigating recombination issues while maintaining codon optimization for specific target organisms. Employing these sensors, the authors successfully engineered AAVs and demonstrated their *in vivo* functionality through fluorescence intensity and lifetime measurements, potentially addressing a challenge in AAV-based *in vivo* imaging. Overall, this manuscript offers a substantial contribution to our understanding of the importance of DNA sequence in developing functional AAVs for *in vivo* imaging. Its implications are noteworthy for Communicational Biology. However, I recommend addressing the following major and minor issues prior to publication:

Major issues:

1. Lines 89–, Fig. 1: The manuscript describes an examination of viral vectors via gel electrophoresis and *in vivo* expression, indicating loss of parts of the FRET sensor gene. It is crucial for the authors to verify and present the original sequence of the FRET sensor gene in the AAV vector plasmid utilized for AAV preparation to confirm its integrity.

We agree with the reviewer that this is important information, and an oversight on our part, due to the fact that recombination was indeed never observed at the plasmid level. We added the corresponding sequencing data as Figure SI3.

2. Lines 221–233: The discussion around the double-acceptor strategy should more thoroughly address the size constraints inherent in AAV engineering. Given the AAV's typical capacity (~4 kb), including essential elements like the promoter, WPRE, and polyA signal, the feasibility of incorporating three fluorescent proteins and an analyte binding domain into a FRET sensor via the ABCD method requires empirical validation through *in vivo* experiments with AAV-encoded sensors.

The total size of the viral cargo (ITR to ITR), for cdATeamDA, was about 4.6 kbp, which is within the tolerated limit (~4.8-5.1 kbp), and the viral production was successful. The analysis of the viral sequences confirmed the presence of a single sequence type of the expected size (Figure SI5). To demonstrate the functionality of the construct, we transduced HEK cells using the produced AAVs and conducted experiments with metabolic blockers, which we included in Figure SI5.

3. Figs. 3g and 4d: The cdFLIIP signal profiles in response to glucose injection raise questions about its functional efficacy as a glucose sensor *in vivo*. The similarity in signal slopes pre- and post-injection does not convincingly support the *in vivo* functionality of the FRET sensor derived from the ABCD method. A critical discussion and appropriate manuscript revision are needed to address this.

The same point was raised by Reviewer 1. We report here the same answer.

We agree with the reviewer that the baseline is not stable. This was already observed in ref.67, and its origin is unclear. To prove that despite this drift the sensor responds to glucose, we performed additional control experiments injecting saline using an identical bolus i.v. injection protocol. We updated the relevant figures and added the following explanation in the text:

Upon glucose injection, cdFLIIP shows ratiometric and lifetime changes compatible with the expected increase in glucose concentration. However, as the curve before injection is not perfectly stable, we performed a control injection using saline. Indeed, a drift of the signal is observed, but the larger change upon glucose injection confirms the sensor's functionality.

An important point here is that we do not want to claim that FLIIP is necessarily the best sensor for glucose in terms of sensitivity or accuracy (which would need to be evaluated in a separate study). The goal of our experiments is rather to demonstrate that the elimination of recombination has restored the functionality of FLIIP as a glucose sensor (and by extension, any sensor with a similar design that would suffer from recombination).

4. Lines 584–586: The citation of previous works (refs. 66 and 67) as demonstrations of the ABCD approach appears misleading, as these references do not explicitly mention this method in the context of FRET sensor design. A revision of this statement is necessary for accuracy.

The mentioned references state that the sensor was codon diversified to avoid recombination, and since those are recent papers that involve several of the authors of this paper and used viral vectors produced by the same viral vector facility involved in this study, we believe it is credible that we used the method described in this work. Nevertheless, we agree with the reviewer that the use of those references might be suboptimal in terms of scientific rigor. At the time of submission, we did not have any other sensor example, but a paper utilizing our method to codon-diversify a sensor for PTEN has been recently published in Nature Methods (<https://www.nature.com/articles/s41592-025-02610-9>). A reference to this work was used instead of ref 66-67.

Minor issues:

1. Ensure consistent use of full terms and abbreviations throughout the manuscript (e.g., fluorescent proteins (FPs), genetically encoded sensors (GESs), calcium (Ca²⁺) etc.).

We thank the reviewer for noticing inconsistencies. We corrected them wherever we found them. For FP vs “fluorescent protein” we used one or the other in some sentences to avoid repetitions and improve readability.

2. The sequence of figure panels in the text appears disorganized, leading to reader confusion. Aligning the figure references in the text with their corresponding panels in a logical sequence is advised.

We apologize the confusion. We have improved the ordering and now, except for Figure 1e (which show data clearly belonging to Figure 1, but cited later in the text in comparison to the results of Figure 2), all figures are cited for the first time in the text according to their alphabetical order.

3. Line 50: “The first FRET sensor” should be more accurately described as “The first FRET GES” to acknowledge the prior existence of chemical dye-based FRET sensors.

We thank the reviewer for this correction.

4. Line 63: Amend “GCamp” to “GCaMP.”

We thank the reviewer for this correction.

5. Fig. 4d: Reorient the Y-axis of the graph to display values from lower (smaller τ) to upper (larger τ) for clarity.

We agree with the reviewer that the inverted axis might appear misleading at first, however we prefer to keep it since the lifetime signal anticorrelates with the glucose concentration, and we believe it is more intuitive to show a curve that grows when glucose concentration grows and viceversa.

6. Lines 505–506: It is generally understood that the analyte binding domain and linkers in a FRET sensor primarily influence the apparent dissociation constant (K_d), rather than the FP domain. If the FP domain does significantly affect K_d , please provide relevant citations for support.

In general, the K_d reflects the relative stability of the sensor in the bound and unbound form. We agree with the reviewer that the K_d of a sensor is primarily influenced by that of its binding unit. Nevertheless, any interaction between amino acids within the sensor contributes to stabilization or destabilization of the bound or unbound form, and thus the side chains on the surface of the FP barrel might influence the K_d as well. The entity of this effect is likely dependent on the specific FP-pair (e.g. the CyPET/YPET pair, that was specifically designed to dimerize efficiently) and sensor. To avoid confusion, we removed the explicit reference to the K_d from the sentence, as the sensor would require re-characterization even in case of no changes to its K_d .

The sentence was modified as:

However, the physicochemical properties of the sensor might be affected, requiring further characterization and re-optimization. In fact, mTFP has been shown to dimerize inefficiently with A. Victoria derived YFPs, leading to sensors with reduced dynamic range.

Changes to main figures

Figure 1:

- (b): Added arrow to point at nucleus
- (f): Improved readability and fixed error with plasmid instead of linear strand in AAV capsid

Figure 3:

- (b): Replaced the Southern Blots (now moved to Figure S15) with previous panel (c), same fixes as in Figure 1f
- (c): Previously panel (f)
- (e): Replaced previous figure based on slopes/ R^2 with new analysis based on coefficient of variation
- (f): Added analysis of nuclear localization
- (g): Included saline curve

Figure 4:

- (c): Added values of coefficient of variation

Reviewer #1 (Remarks to the Author):

In this revised manuscript, Dernic et al. have performed additional experiments, new analysis and textual edits that have addressed my prior comments. I support the publication of this manuscript. However, there are still a couple of minor points that I would like them to be addressed.

We thank the reviewer for his/her positive comments. We address below the remaining points.

1) The FLIIP image in Fig. 1b is a duplication of that in Fig. 3d. Can one of them be replaced?

We removed the image of the recombining sensor from Figure 3d.

2) “glc iv” in Figure 3g is probably better changed to “iv” since saline injection is also used.

We thank the reviewer for catching this typo. We fixed it.

3) The n for some experiments is missing. E.g., in legend of Fig. 4d, please provide number of neurons/mice.

We added the missing information in the legends.

Reviewer #3 (Remarks to the Author):

The authors have satisfactorily addressed all of the concerns raised in the previous review. The revised manuscript is now suitable for publication in its current form.

We thank the reviewer for the appreciation.

Reviewer #4 (Remarks to the Author):

I had the chance to look at the manuscript from Dernic et al only as a revised manuscript, and not at the time of the first submission.

The new Figure 3 and the Tables, together with additions to the narrative, greatly improve the quality of the paper. The proposed strategy is clever and holds promises for biosensor optimization on a wider scale.

We thank the reviewer for the positive evaluation of our work. We address below to the remaining points.

There are a few points that remain to be solved:

1. Despite the answer provided by the authors, I also believe that there are long stretches along the diagonal of ECFP/cdCitrine (M20) plot in Fig. 2f although there is no red line highlighting them. Can the authors comment on this? The diagonal pattern is evident by eye.

The color of the lines for ECFP/cdCitrine (M20) in Fig. 2f reflects the distribution of lengths in the rightmost column of Fig. 2e, with the longest identical stretches being 9 nucleotides. On the other hand, the ECFP/Citrine pair showcases stretches of up to about 200 nucleotides. To allow the visualization of differences between short sequences, the color scale maximum was chosen as 20 nucleotides, and thus all the long stretches for ECFP/Citrine appear as red. In contrast, for ECFP/cdCitrine (M20) the blue-green colors are consistent with lengths of less than 10 nucleotides. The fact that a “diagonal pattern” can be extrapolated by eye is consistent with the fact that the “long identical stretches” were divided in “short identical stretches” separated by diversified stretches. This is similar to a dashed line, which is still recognizable as a line by eye, even though it is not continuous. Also, due to the resolution of the image, it is sometimes possible that two stretches appear continuous (e.g. in proximity of the green stretch in the bottom right corner). In this case, the color is a more reliable visual indication.

2. I appreciate the reference to the recently published Nat Methods paper, which shows a PTEN biosensor built by taking advantage of the strategy implemented by the authors. However, this does not fully answer the issue of how generalized this strategy can be. The Nat Methods paper reports on a novel sensor, and unless I’m mistaken, there is no comparison between an unmodified and a cd-modified version of the sensor. Concerning ATeam, I acknowledge that the only fluorescent product upon recombination is the fluorescent one. However, can codon diversification further stabilize the readout of the sensor and make it more robust in cells where expression levels of the sensor are naturally low, for instance? This would substantiate that “losing” a fraction of the unmodified ATeam would be detrimental to the S/N ratio and therefore, to correctly estimate FRET. In this case, expressing cdATeam would be a clear advantage to improve FRET detection. Overall, I believe that a second example highlighting the breadth of the codon optimization strategy would be beneficial to generalize the strategy.

Regarding ATeam, the reviewer is correct in pointing out that, given an identical injected titer, a recombining sensor would lead to a lower expression, hence a lower SNR and/or a

larger contamination by autofluorescence. On the other hand, it could be argued that increasing the injected titer (when possible) might compensate for the problem, at the expense of a larger risk of immune response and/or increase of risk of unknown effects due to the large amount of “non-fluorescent junk” expressed by the cell. All those effects though are extremely difficult to prove, and we don’t think it would add much to the fact that eliminating recombination would anyways be the safest option.

Regarding the PTEN sensor, we now included a new Supplementary Figure to show a comparison of the sensor before and after diversification, clearly showing the improvement. The following paragraph was added to the discussion:

A further demonstration of the utility of our methodology has been recently provided by the diversification of the phosphatase sensor G-PTEN, a FLIM-FRET sensor that features a dark acceptor (Supplementary Figure 8). In this case, the recombination effect is harder to recognize, as its product (an EGFP-like FP mix) has a similar lifetime to the low-FRET form of the sensor. However, the impact of recombination is clearly visible in the high-FRET form, which displays a much-reduced variability upon diversification.

3. Fig. 3g and 4d. Why is the FLIIP–saline condition missing? Do we see a difference between glucose and saline with the original sensor? If that were the case, is the delta of the response between saline and glucose maintained when FLIIP and cdFLIIP are compared? I believe this control condition is necessary to fully appreciate the improved response of the cdFLIIP construct.

We considered performing a control experiment also for the FLIIP sensor, however we realized that this would not have been a determinant factor in assessing whether cdFLIIP had a larger glucose sensitivity than FLIIP. In fact, even in the very unlikely case in which in the FLIIP experiments there is a downward trend that exactly counteracts an upward response at all times (including a necessary change of slope of the trend corresponding with the glucose injection), the result is a “sensor” that upon injection of glucose does not produce a response. Thus, FLIIP is inferior to cdFLIIP in responding to the same glucose protocol. Furthermore, even if a delta was observed, this might give the false impression that the sensor could be somehow usable, while its downfalls are clearly demonstrated by its false report of a large cell-to-cell variability (Figure 3d,e). This is fully consistent with the genetic analysis of the viral preparations and the nuclear localization of the signal, which show that the fluorescence originates largely from single FPs, that are incapable of sensing glucose as they lack the binding unit.

Changes to main figures

Figure 3:

- Removed FLIIP from scheme b
- Replaced “glc i.v.” with “i.v. injection”

Figure4:

- Replaced “glc i.v.” with “i.v. injection”

Dernic et al. introduce a novel strategy for the sequence design of genetically encoded Förster Resonance Energy Transfer (FRET) sensors for engineering functional adeno-associated virus (AAV) *in vivo*. This approach, designated ABCD, computationally designs diverse DNA sequences for multiple fluorescent proteins in the sensor, effectively mitigating recombination issues while maintaining codon optimization for specific target organisms. Employing these sensors, the authors successfully engineered AAVs and demonstrated their *in vivo* functionality through fluorescence intensity and lifetime measurements, potentially addressing a challenge in AAV-based *in vivo* imaging.

Overall, this manuscript offers a substantial contribution to our understanding of the importance of DNA sequence in developing functional AAVs for *in vivo* imaging. Its implications are noteworthy for *Communicational Biology*. However, I recommend addressing the following major and minor issues prior to publication:

Major issues:

1. Lines 89–, Fig. 1: The manuscript describes an examination of viral vectors via gel electrophoresis and *in vivo* expression, indicating loss of parts of the FRET sensor gene. It is crucial for the authors to verify and present the original sequence of the FRET sensor gene in the AAV vector plasmid utilized for AAV preparation to confirm its integrity.
2. Lines 221–233: The discussion around the double-acceptor strategy should more thoroughly address the size constraints inherent in AAV engineering. Given the AAV's typical capacity (~4 kb), including essential elements like the promoter, WPRE, and polyA signal, the feasibility of incorporating three fluorescent proteins and an analyte binding domain into a FRET sensor via the ABCD method requires empirical validation through *in vivo* experiments with AAV-encoded sensors.
3. Figs. 3g and 4d: The cdFLIIP signal profiles in response to glucose injection raise questions about its functional efficacy as a glucose sensor *in vivo*. The similarity in signal slopes pre- and post-injection does not convincingly support the *in vivo* functionality of the FRET sensor derived from the ABCD method. A critical discussion and appropriate manuscript revision are needed to address this.
4. Lines 584–586: The citation of previous works (refs. 66 and 67) as demonstrations of the ABCD approach appears misleading, as these references do not explicitly mention this method in the context of FRET sensor design. A revision of this statement is necessary for accuracy.

Minor issues:

1. Ensure consistent use of full terms and abbreviations throughout the manuscript (e.g., fluorescent proteins (FPs), genetically encoded sensors (GESs), calcium (Ca^{2+}) etc.).
2. The sequence of figure panels in the text appears disorganized, leading to reader confusion. Aligning the figure references in the text with their corresponding panels in a logical sequence is advised.
3. Line 50: “The first FRET sensor” should be more accurately described as “The first FRET GES” to acknowledge the prior existence of chemical dye-based FRET sensors.
4. Line 63: Amend “GCamp” to “GCaMP.”
5. Fig. 4d: Reorient the Y-axis of the graph to display values from lower (smaller τ) to upper (larger τ) for clarity.
6. Lines 505–506: It is generally understood that the analyte binding domain and linkers in a FRET sensor primarily influence the apparent dissociation constant (K_d), rather than the FP domain. If the FP domain does significantly affect K_d , please provide relevant citations for support.